# Comparison of soil production, chemical weathering, and physical erosion rates along a climate and ecological gradient (Chile) to global observations

Mirjam Schaller[1], Todd A. Ehlers[1]

[1]University of Tuebingen, Department of Geosciences, Schnarrenbergstrasse 94-96, 72076 Tuebingen, Germany

*Correspondence to*: Mirjam Schaller (mirjam.schaller@uni-tuebingen.de)

**Abstract**

Weathering of bedrock to produce regolith is essential for sustaining life on Earth and global biogeochemical cycles. The rate of this process is influenced not only by tectonics, but also by climate and biota. We present new data of soil production,

chemical weathering, and physical erosion rates from the large climate and ecological gradient of the Chilean Coastal Cordillera (26° to 38°S). Four Chilean study areas are investigated and span (from north to south): arid (Pan de Azúcar), semi-arid (Santa Gracia), mediterranean (La Campana), and temperate humid (Nahuelbuta) climate zones. Observed soil production rates in granitoid soil-mantled hillslopes range from ~7 to 290 t/(km$^2$ yr) and are lowest in the sparsely vegetated and arid north and highest in the Mediterranean setting. Calculated chemical weathering rates range from zero in the arid north to a

high of 211 t/(km$^2$ yr) in the mediterranean zone. Chemical weathering rates are moderate in the semi-arid and temperate humid zones (~20 to 50 t/(km$^2$ yr)). Similarly, physical erosion rates are lowest in the arid zone (~11 t/(km$^2$ yr)) and highest in the mediterranean climate zone (~91 t/(km$^2$ yr)). The contribution of chemical weathering to total denudation rates is lower in the arid north than further south. However, due to heterogeneities in lithologies and Zr concentrations, reported chemical weathering rates and chemical depletion fractions are affected by large uncertainties. Comparison of Chilean results to

published global data collected from hillslope settings underlain by granitoid lithologies document similar patterns in soil production, chemical weathering, and total denudation rates for varying mean annual precipitation and vegetation cover amounts. We discuss the Chilean and global data in the light of contending model frameworks in the literature and find that observed variations in soil production rates bear the closest resemblance to models explicitly accounting for variations in soil thickness and biomass.

## 1 Introduction

Regolith forms through the weathering of rock near the Earth's surface and contains a mobile soil[1] that overlies an immobile saprolite (weathered bedrock) (e.g., Heimsath et al., 1997; Riebe and Granger, 2013). Biotic and abiotically driven chemical weathering and physical erosion of the mobile soil layer results in the continuous supply of weathered bedrock towards the surface and replenishes nutrients for biota (e.g., Amundson et al., 2007). Soil production from weathered bedrock in a soil-mantled hillslope occurs through complex interactions between tectonics (which influences the slope of topography and therefore denudation), rock type, climate (specifically precipitation and temperature), and biota (for instance vegetation). All of these soil production processes are active over extended (millennial and greater) timescales (e.g., Mishra et al., 2019). Disentangling the interactions between tectonics, climate, and biota on soil production has proved challenging and has previously been approached in study areas with diverse geologic, climate, and vegetation histories through the determination of soil production rates (e.g., Dixon et al., 2009; Dixon and von Blanckenburg, 2012; Heimsath et al., 2012; Larsen et al., 2014). The term soil production rate (SPR) is used here as a general term for the production rate of bedrock or saprolite into soil. Studies investigating SPRs typically operate under the simplifying assumption of steady-state conditions. Under steady-state conditions, SPRs are considered equal to surface denudation rates, which include mass loss through chemical weathering and physical erosion in soil and saprolite (e.g., Dixon et al., 2009). In this study, we build upon previous work and quantify soil production, chemical weathering, and physical erosion rates across a diverse climate and ecological gradient. Our emphasis is on observations from soil-mantled hillslopes in both granitoid settings in the Coastal Cordillera of Chile, and elsewhere globally.

Over the past decades, a growing number of studies have addressed the relationship between chemical weathering and physical erosion. A recent review by Riebe et al., (2017) provides a useful starting point for understanding different perspectives. For chemical weathering, two end-member perspectives have been proposed and suggest that chemical weathering rates can be either supply- or kinetically-limited (e.g., Riebe et al., 2004a; West et al., 2005). End-member models considered for physical erosion rates are framed around being either transport- or weathering-limited (e.g., Gilbert, 1877; Carson and Kirkby, 1972 amongst many others). The previous end-member models were formulated without explicit consideration of how biota influence soil production, chemical weathering, and physical erosion. Complimentary to the previous work, other studies have focused on identifying the role of biota in chemical and physical processes near the Earth's surface (e.g., see review by Amundson et al., 2007; Starke et al., 2020; Oeser and von Blanckenburg, 2020).

---

[1] The terms regolith, soil, and saprolite used in this study follow the notation of Riebe and Granger (2013). This notation is not consistent with the one used in soil sciences (IUSS Working Group WRB, 2015) where the terms soil/regolith, pedolith, and saprolith would be used, respectively.

In this study, we focus on observations from soil-mantled hillslopes underlain by granitoid lithologies. Our choice in doing this minimizes potential lithologic variations influencing results, and also avoids discrepancies between hillslope vs. catchment

average river load approaches due to different timescales of observation or stoichiometry effects (see discussion in West et al., 2005). With respect to soil-mantled hillslopes, previous work by Riebe et al. (2001) investigated chemical weathering rates based on the combination of cosmogenic nuclides measured in river load and immobile element concentrations (e.g., Zr or Ti) collected from soil, saprolite, and bedrock exposed in pedons. In this approach, the enrichment of immobile elements in soil and saprolite (relative to bedrock) was used to separate denudation rates into chemical weathering and physical erosion rates

(see also Riebe and Granger, 2013). In subsequent work, Riebe et al. (2004a) investigated different granitoid settings around the world, spanning mean annual precipitation (MAP) rates between 220 to 4200 mm/yr and mean annual temperatures (MAT) between 2 to 25ºC. They observed that the highest chemical weathering rates occur in areas with high denudation rates. Other studies have also recognized the importance of climate and vegetation on weathering rates. For example, the influence of climate on SPRs and chemical weathering rates has been documented from samples collected along altitudinal transects (e.g.,

Riebe et al, 2004b; Dixon et al., 2009; Ferrier et al., 2012). More specifically, a decrease in chemical weathering rates was observed with increasing altitude (and hence decreasing temperature) and was attributed to changes in both snow depth and vegetation cover (Riebe et al., 2004b). In a different location (Dixon et al., 2009), an altitudinal transect documented a peak in chemical weathering rates that correlated with physical erosion rates in middle elevation positions. Dixon et al. (2009) attributed this result to climate variations affecting vegetation along the transect. In contrast to the previous studies, chemical

weathering rates from two transects collected below tree line elevation were observed to be insensitive to regolith temperature differences as well as regolith moisture (Ferrier et al., 2012). Ferrier et al. (2012) concluded that there was no significant change in weathering rates observed due to the lack of biotic changes along the transects. Previous work has also identified that the contribution of chemical weathering rates to total denudation rates appears to remain constant for different denudation rates despite changes in MAP (e.g., Riebe et al., 2004a; Larsen et al., 2014). Taken together, the previous work suggests a

sensitivity of chemical weathering rates to precipitation, temperature, and vegetation cover (Riebe et al., 2004b; Dixon et al., 2009), and (inferred) sensitivity to biotic changes (Ferrier et al., 2012), and an insensitivity to regolith temperatures and moisture. Taken together, the previous studies contain conflicting results and highlight uncertainty in our current knowledge. Building upon the previous work, Oeser et al. (2018) and Schaller et al., (2018) investigated four study areas located along the extreme climate and ecological gradient of the Chilean Coastal Cordillera (26° – 38°S), which are part of the "EarthShape"

German-Chilean priority research program (www.earthshape.net). Their work documented that the total denudation rates first increase from north to south and then decrease (or possibly stay the same) further south despite increasing MAP rates and vegetation cover (Oeser et al., 2018, Schaller et al., 2018). In these Chilean study areas, the contribution of chemical weathering to total denudation rates was variable and almost zero in the arid north, ~50% in the semi-arid and mediterranean settings, but appeared reduced or potentially equal (to the mediterranean setting) to the temperate climate of the southernmost

study area (e.g., Oeser et al., 2018). However, due to regolith compositional heterogeneities in the southernmost study area, a high variability in Zr concentrations was observed and calculated chemical weathering rates were not considered reliable.

Finally, recent work by Oeser and von Blankenburg (2020) conducted in the same four Chilean study areas used $^{87}Sr/^{86}Sr$ isotope and other elemental variations measured in bedrock, regolith, soil, and vegetation to infer nutrient uptake vs. recycling processes. They found that the weathering degree and rate do not increase from the arid north to south with increased net primary productivity, but rather higher biomass nutrient demands in the south are most likely accommodated through faster nutrient recycling in this ecosystem.

In this study, we build upon the previous work in two ways. First, we present 11 new cosmogenic nuclide measurements collected from four study areas along ~1,300 km of the extreme climate and ecological gradient of the slowly eroding Chilean Coastal Cordillera (Fig. 1). These samples come from new pedon locations not previously investigated by Schaller et al. (2018), but located along the same hillslopes and with major element measurements available from previous work by Oeser et al, (2018). Given the contrasting results of previous studies (e.g., Riebe et al., 2004b; Dixon et al., 2009; and Ferrier et al., 2012) our aim with these new samples was to verify if our previous observations (Oeser et al., 2018) of latitudinal changes in soil production, chemical weathering, and physical erosion rates are valid for pedons located in different (i.e., top- and toe-slope) hillslope positions than the previously investigated mid-slope 'only' positions considered in Oeser et al. (2018). Second, equipped with the new results from the Chilean transect, we compare our Chilean analysis to globally similar data sets and sampling approaches from granitoid hillslopes. This comparison is done to evaluate the similarity or dissimilarity in observations between areas. Together, these two approaches provide a holistic picture of if any non-linear relationship(s) in soil production, chemical weathering, and physical erosion rates exist across different climate and vegetation settings. To address the previous two aims, our efforts are focused on evaluating two hypotheses. These include: 1) soil production as well as chemical weathering rates increase with increasing MAP rates. This hypothesis stems from the model considerations of Norton et al. (2014) and a data compilation of Perron (2017) as well as White and Blum (1995); and 2) the contribution of chemical weathering to total denudation is constant over a climate gradient. Hypothesis 2 is supported by previous work of Riebe et al. (2004a) and Larsen et al. (2014). If this hypothesis is true, the proposed relationship could be evident from the extended Chilean transect data presented here. We note that throughout this manuscript, our calculation of chemical weathering and physical erosion rates makes use of the geochemical data reported by Oeser et al. (2018), but follows the approach, notation, and methods described in Dixon et al. (2009).

## 2 Chilean study areas

The four study areas are located in the Chilean Coastal Cordillera (~26° to 38° S, Fig. 1; Table 1) and span (from north to south) climate and biogeographic zones ranging from arid to humid. Each area contains primarily granitoid lithologies (mostly granodiorite, tonalite, quartz diorite) and to a lesser degree gabbro in the Santa Gracia area (Oeser et al., 2018). The Coastal Cordillera lies along the west coast of Chile, where the neighboring plate boundary is similar along strike underneath the Coastal Cordillera due to subduction of the Nazca Plate. The study areas were selected to keep differences in lithology and tectonic setting to a minimum and to enhance the signal of variable climate and vegetation on denudation rates.

In the following, we present the primary characteristics of the Chilean study areas that contain pedons in both south- and north-facing slopes. In all the study areas, no signs of glaciation are reported in previous literature or our own field observations. One caveat worth discussion is that the data sets considered in this study are sensitive to different time scales. For example, cosmogenic nuclide-derived soil production rates typically integrate over timescales of $\sim 10^2\text{-}10^6$ years, whereas vegetation and climate change can occur over similar and much shorter timescales. The comparison of modern climate and vegetation metrics to cosmogenic nuclide-derived rates sensitive to these different timescales can introduce uncertainty into any interpretation. However, recent work has quantified climate and vegetation changes in the four study areas since the Last Glacial Maximum (LGM) and up to the Pliocene (e.g., Mutz et al., 2018; Werner et al., 2018). While LGM climate was wetter and cooler, the climate and vegetation gradients present today also existed in the past (Starke et al., 2020; Mutz et al., 2018; Mutz and Ehlers, 2019), thereby diminishing concerns that modern climate and vegetation gradients are a poor representation of past conditions that cosmogenic nuclide-derived rates may be sensitive to. With this potential complication in mind, we proceed with presenting the present-day characteristics of each study area. Numbers reported in sections 2.1 to 2.4 are from previous work by Bernhard et al. (2018), Oeser et al. (2018), Schmid et al., (2018), and Schaller et al. (2018). The combined thickness of A- and B-horizons is considered as soil thickness (see Table S1 in Oeser et a., 2018). The reported clay content, pH, and bulk density are the pedon averages of each study area (see Table 3 in Bernhard et al., 2018). The chemical index of alteration (CIA; after Nesbitt and Young, 1982) for bedrock is a study area average whereas the CIA for regolith is reported for specific horizons (for more details see Table S5 in Oeser et al., 2018). The cosmogenic nuclide-derived denudation rates are reported for south- and north-facing mid-slope positions (see Oeser et al., 2018 and Table S6 in there).

## 2.1 Pan de Azúcar

In the northernmost study area Pan de Azúcar (Fig. 1; Fig. S1) MAP and MAT are 8 mm/yr and 21.1 °C (Karger et al., 2017). The very sparse higher desert vegetation covers 2% of the land surface. The Regosol containing pedogenic gypsum has soil thicknesses of 20 to 25 cm. The regolith has a clay content of 13.8 ±3.9%, a bulk density of 1.3 ±0.1 g/cm³, and a pH of 8.1 ±0.1. The average CIA in bedrock is 55 whereas the CIA in regolith horizons can be as low as 31 due to atmospheric input. Cosmogenic nuclide-derived denudation rates from south- and north-facing mid-slope positions are 11.0 ±0.7 and 8.2 ±0.5 t/(km² yr) and dominated by physical erosion whereas chemical weathering is insignificant.

## 2.2 Santa Gracia

In Santa Gracia (Fig. 1; Fig. S1), MAP and MAT are 97 mm/yr and 17.7 °C, respectively (Karger et al., 2017). The soil horizons of this Cambisol are 30 to 55 cm thick. The clay content of the regolith is 11.1 ±4.9%, the regolith bulk density is 1.5 ±0.0 g/cm³, and the pH is 6.3 ±0.3. The average CIA in the bedrock is ~43, which is similar to other granitoid lithologies (i.e., 45 to 55; Nesbitt and Young, 1982) but lower than the other three study areas (52 to 55). The CIA in regolith horizons ranges from 49 to 52, indicating regolith weathering. The cosmogenic nuclide-derived denudation rates in the south- and

north-facing mid-slope positions are 22.4 ±1.5 and 15.9 ±0.9 t/(km² yr). Chemical weathering rates are comparable to physical erosion rates.

## 2.3 La Campana

A mediterranean dry forest grows in La Campana (Fig. 1; Fig. S1) where MAP and MAT are 307 mm/yr and 14.1 °C, respectively (Karger et al., 2017). This forest covers 84% of the land surface. The Cambisol contains 35 to 60 cm thick soil
horizons and a clay content of 10.5 ±1.6%. The regolith bulk density is 1.3 ±0.2 g/cm³ whereas the pH is 5.4 ±0.3. The average CIA of the bedrock is 52 whereas the CIA of regolith horizons ranges between 50 to 58. Values of the cosmogenic nuclide-derived denudation rates are 53.7 ±3.4 and 69.2 ±4.6 t/(km² yr) for the south- and north-facing mid-slope positions. Reported chemical weathering rates are comparable to physical erosion rates.

## 2.2 Nahuelbuta

In the southernmost study area of Nahuelbuta (Fig. 1; Fig. S1), the MAP and MAT are 1,479 mm/yr and 6.1 °C (Karger et al., 2017), respectively. The area contains a temperate mixed broadleaved-coniferous forest. Vegetation cover is as high as 95%. The regolith type is an Umbrisol with soil horizons as thick as 60 to 90 cm and a clay content of 26.2 ±2.6% in the regolith. Regolith bulk density and pH are 0.8 ±0.1 g/cm³ and 4.3 ±0.2, respectively. The average CIA of bedrock is 54, whereas values in regolith horizons are as high as 75, indicating substantial weathering. Cosmogenic nuclide-derived denudation rates are
47.5 ±3.0 and 17.7 ±1.1 t/(km² yr) in the south- and north-facing mid-slope positions. Chemical weathering rates appear to be lower than physical erosion rates.

## 3 Material and methods

Soil production, chemical weathering, and physical erosion rates were determined, or recalculated, for top-, mid-, and toe-slope positions in the south-facing slopes, whereas only the mid-slope is recalculated for the north-facing slopes. Cosmogenic
nuclide concentrations of saprolite from 8 mid-slope positions are from Schaller et al. (2018) whereas 11 new top- and toe-slope position samples are presented here. Material analyzed for [10]Be concentrations to calculate SPRs are collected from the top of the saprolite made accessible by pedon excavation (e.g., www.earthshape.net; Bernhard et al., 2018; Oeser et al., 2018). The top of saprolite is considered to be the first encounter of in situ-weathered bedrock represented by the C-horizon. This sampling strategy is a common approach for calculation of soil production rates from cosmogenic nuclide measured in pedons
(e.g., Dixon et al., 2009). Representative photographs of this horizon from the Chilean study areas are available in Oeser et al., (2018: Figures 3 to 6). In our study, the depths sampled in each pit are shown in Table S1. Saprolite material was crushed and sieved into a 0.25–1.0 mm fraction. Further cleaning of sample material and extraction of Be followed the methods as described in Schaller et al. (2018). For previous, and all but one new, sample locations the chemical weathering and physical

erosion rates were calculated from the SPRs and major and trace element concentrations in soil, saprolite, and bedrock. Major and trace element concentrations were used from Oeser et al. (2018) to calculate chemical weathering. To summarize, we combine 11 new, with 8 previously published, cosmogenic nuclide concentrations to present 19 newly calculated SPRs and 18 newly calculated chemical weathering and physical erosion rates.

### 3.1 Soil production rates

SPRs from Chilean soil-mantled hillslopes (Table S1) were calculated as follows: The in situ-produced [10]Be concentration of quartz contains information about the regolith or SPR (e.g., Heimsath et al., 1997). The nuclide concentration C (atoms/$g_{(qtz)}$) of the uppermost saprolite sample at the sample location is a function of:

$$C = \sum_1^3 \frac{P_n(h,\theta)}{(\lambda + \frac{SPR}{\Lambda_n})} \tag{1},$$

where $P_n$ is the nuclide production rate (atoms/($g_{(qtz)}$ yr)) at sample depth h (cm) corrected for shielding $\Theta$ (unitless) by nucleonic, stopped muonic, and fast muonic production, respectively. No shielding by topography (less than 1%), snow, or vegetation has been taken into account). The SPR is the soil production rate (g/(cm$^2$ yr)), $\lambda$ is the decay constant (/yr), and $\Lambda_n$ are the mean nucleonic, stopped muonic, and fast muonic attenuation lengths of 157, 1500, and 4320 g/cm$^2$, respectively (Braucher et al., 2011). The nucleonic, stopped muonic, and fast muonic production rates at sea level high latitude of 3.92, 0.012, and 0.039 atoms/($g_{(qtz)}$ yr) (Borchers et al., 2016) were scaled to the sample location based on Marrero et al. (2016).

### 3.2 Chemical weathering and physical erosion rates

The calculation of chemical weathering and physical erosion rates for the Chilean study areas follows the equations in Dixon et al. (2009). The sum of all mass loss $D_{total}$ (t/(km$^2$ yr)) is given by:

$$D_{total} = SPR + W_{sap} = E_{soil} + W_{soil} + W_{sap} \tag{2},$$

where SPR is the known soil production rate (t/(km$^2$ yr)), $W_{sap}$ is the chemical weathering rate for saprolite (t/(km$^2$ yr)), $W_{soil}$ the chemical weathering rate for soil (t/(km$^2$ yr)), and $E_{soil}$ is the physical erosion rate for soil (t/(km$^2$ yr)). Following Riebe et al. (2004a) and Dixon et al. (2009), $W_{sap}$ can be expressed with the help of immobile elements in saprolite and bedrock as:

$$W_{sap} = SPR(\frac{Zr_{sap}}{Zr_{rock}} - 1) \tag{3},$$

where $Zr_{sap}$ and $Zr_{rock}$ are the concentration of Zr in saprolite and bedrock, respectively. In contrast, $W_{soil}$ can be expressed by:

$$W_{soil} = SPR(1 - \frac{Zr_{sap}}{Zr_{soil}}) \tag{4},$$

where $Zr_{soil}$ is the average Zr concentration for soil samples from the pedon. Similarly, $Zr_{sap}$ is the average Zr concentration of the saprolite samples from the pedon. $Zr_{rock}$ is based on the average of all bedrock samples collected in one specific study area (see Table S2 based on Oeser et al., 2018). The chemical depletion fraction $CDF_{total}$ is determined by the fraction of the total chemical weathering rate $W_{total}$ (as the sum of $W_{soil}$ and $W_{sap}$) and the total denudation rate $D_{total}$ (Tables S3 and S4). As all results presented below are based on the notation of Dixon et al. (2009), the total denudation rates include the contribution

of weathering in the saprolite. We note that our approach for calculating total denudation rates is slightly different from that of Oeser et al. (2018) who did not account for weathering in the saprolite and whose numbers can be considered as minimum denudation rate estimates. As a result, the SPRs reported here and the total denudation rates in Oeser et al, (2018) are the same expect for La Campana and Nahuelbuta where Oeser et al. (2018) corrected rates for vegetation cover.

## 3.3 Global data compilation, correlation, and model simulations

The hypotheses addressed in this study relate to the relationships between soil production, chemical weathering, physical erosion, and factors such as precipitation, slope, and vegetation cover. To evaluate these hypotheses, we present an analysis for both the new Chilean transect data and also a global compilation of previously published data from similar granitoid sample locations as the Chilean data. More specifically, SPRs from the Chilean study areas were compared to previously published SPRs derived from granitoid (Table 2 and S5) and non-granitoid (Table S6) soil-mantled hillslopes from around the world.

We note that any comparison to global inventories can be prone to an oversimplification of the natural variability between locations and relevant processes active therein. Indeed, many previous studies investigating silicate rock weathering and denudation relationships have had to make important assumptions in their data compilations and comparisons (e.g., see discussions in West et al., 2005; Ferrier et al., 2016; and Riebe et al., 2017). To avoid oversimplifying our comparison of the Chilean data to other globally distributed studies, we have carefully selected previous studies reporting data most akin to our sampling and analysis approach. For example, global sites we directly compare our results to were taken from soil-mantled hillslope measurements rather than catchment average estimates from river load, and also locations with granitoid lithologies underlying the hillslopes to minimize lithologic variability effects. Although we include settings with non-granitoid hillslopes, we do this for completeness to highlight how lithologic variations might induce variability, but we avoid interpreting differences between these areas and our Chilean results. For all study locations considered (Chilean or global) we use the same, well-established, global data sets for leaf area index LAI (TERRA/MODIS; Knyazikhin et al., 1999), and precipitation and temperature (CHELSA, Karger et al., 2017). Global LAI and climate data sets were used instead of region-specific data sets to avoid inherent differences and biases between studies due to varying methods associated with their data processing, timescales of observations, and different observational data sets used between studies. The global data sets used here for LAI and climate therefore benefit from having the same, consistent, processing of data, but suffer from having a coarser resolution than local to regional based studies. Topographic information (e.g., hillslope angle) was taken from the original publications when possible as the local measurements are assumed to be more accurate than slope angles from DEMs which could introduce biases due to the DEM resolution considered. Finally, one potential bias present in our comparison is that SPRs reported here and in published data are often measured in areas where soils are thick. The potential bias introduced by this is that these areas may be supply-limited. There is no easy way to correct for this bias given available data, but we mention it as a potential caveat associated with this (and others) analysis. However, we do note that the global data available for comparison (n=303) span both a wide range of MAP and SPR often times within the same geographic location such that this potential caveat may not be of concern. Thus, the compilation presented here of 'pruned' global observations aims at the fairest comparison possible

between different areas, but does so at the risk of excluding some readers 'favorite' study areas that don't meet the objective application of criteria described here.

For the Chilean and global data sets our analyses were conducted in two ways. First, for the globally distributed granitoid sample locations, a range of topographic, climate, and vegetation parameters were compiled (Table S5). Soil production rates and some parameters were also compiled for non-granitoid sample locations (Table S6). The SPRs and parameters for granitoid sample locations were analyzed with a Pearson linear correlation analysis (Table S7). Furthermore, chemical weathering and physical erosion rates determined in granitoid soil-mantled hillslopes around the world were compiled (Tables 2 and S8). The previous climate and vegetation parameters were compared to chemical weathering and physical erosion rates also using a Pearson correlation analysis (Table S9).

In addition (second), we compare observations from both the Chilean and global data to previously proposed theoretical/modelling approaches. This comparison is done to evaluate the performance of different models and identify what processes (rather than just correlations) are supportive of structure within the data sets. In particular, we compare results to the models of: (1) Norton et al. (2014) which predicts the maximum SPR for different MAP and MAT; (2) a modified version of the Norton et al. (2014) approach that also accounts for the effect of variable soil thickness with changing MAP on SPR; and (3) Pelak et al. (2016) who predicted SPR as a function of biomass density and soil depth. The details of each approach and governing equations are provided in the Supplementary Material S1. Finally, for completeness we also compared results (presented in Supplemental Material S1) to the effective energy and mass transfer (EEMT) approach of Pelletier and Rasmussen (2009). This later approach accounts for SPR changes as a function of MAT and MAP taking into account the influence of the biotic control on EEMT. However, the results are not presented in the main text because this model did not provide a satisfactory fit to the hillslope observations, which could be due to (in part) that the model was applied to measurements on river load. Nevertheless, we include it in the Supplementary Material for the curious reader.

## 4 Results

In this section, results for soil production, chemical weathering, and physical erosion rates are presented for the new and previously published cosmogenic nuclide concentrations from the Chilean study areas (Fig. 2; Tables S1, S3, and S4). Results are given for each study area starting in the arid north and progressing to the south. The total denudation rates ($D_{total}$) presented below are the composite of the total chemical weathering rate ($W_{total}$) and the physical erosion rate ($E_{soil}$) and based on the calculated SPRs and the Zr concentrations in rock, saprolite, and soil (Table S2). The previously published total denudation rates (Oeser et al., 2018) were recalculated to account for chemical weathering of saprolite (see methods). Because the observations presented in our global compilation were previously published (see references in Table 2), we do not present the compilation in this section but rather in section 5 (Discussion) where it is integrated and discussed in the context of our Chilean observations. In addition, correlations of SPRs as well as chemical weathering and physical erosion rates with parameters are also shown in section 5.

## 4. 1 Pan de Azúcar

The in situ-produced $^{10}$Be concentrations in quartz range from (2.58 ±0.08) $10^5$ atoms/g$_{(qtz)}$ to (5.08 ±0.16) $10^5$ atoms/g$_{(qtz)}$ (Table S1). The south-facing slope of Pan de Azúcar has SPRs that increase from 7.7 ±0.5 t/(km$^2$ yr) in the top-slope position to 16.8 ±1.0 t/(km$^2$ yr) in the toe-slope (Fig. 2A; Table S1). The north-facing mid-slope position has a lower SPR than the south-facing mid-slope position. The average Zr concentration of bedrock in Pan de Azúcar is 206 ±8 ppm, whereas Zr concentrations in soil and saprolite are between 184 ±14 and 272 ±75 ppm (Table S2). The averaged Zr$_{soil}$/Zr$_{rock}$ ratio in Pan

de Azúcar is 1.07 ±0.09 (Table S3) indicating no enrichment of the immobile element Zr from rock to soil. Chemical weathering rates are low <1.7 ±0.8 t/(km$^2$ yr); Fig. 2B; Table S3) and with an average value of 0.3 ±0.3 t/(km$^2$ yr). The average total denudation rate of 11.0 ±0.24 t/(km$^2$ yr) is dominated by the physical erosion rate with an average value for the study location of 10.7 ±2.6 t/(km$^2$ yr). This results in an average CDF$_{total}$ of 0.05 ±0.08 (Table S4).

## 4. 2 Santa Gracia

In Santa Gracia, $^{10}$Be concentrations in quartz range from (1.75 ±0.06) $10^5$ atoms/g$_{(qtz)}$ and (4.19 ±0.23) $10^5$ atoms/g$_{(qtz)}$ (Table S1). In the south-facing slope of Santa Gracia, where six samples were collected from top- to toe-slope positions (Tables 1 and S1), SPRs generally increase downslope from 17.8 ±1.3 to 28.0 ±1.7 t/(km$^2$ yr). The SPR in the north-facing mid-slope position is lower than the equivalent mid-slope position in the south-facing slope (Fig. 2A; Table S1). The Zr$_{soil}$/Zr$_{rock}$ ratios are around 2 but subject to a large uncertainty due to inhomogeneity in the Zr$_{rock}$ concentration (Table S2). Nevertheless, the

averaged ratio of Zr$_{soil}$/Zr$_{rock}$ is 2.10 ±0.08 indicating enrichment of the immobile element Zr from rock to soil. An estimated 80% of the weathering is located in the saprolite (Table S3). Total chemical weathering rates range from 11.0 ±5.4 t/(km$^2$ yr) to 30.4 ±16.0 t/(km$^2$ yr) (Fig. 2B). Higher chemical weathering rates are calculated for weathering in saprolite than in soil (Fig. S2; Table S3). Total chemical weathering and physical erosion rates contribute equally to the total denudation rate (Table S4). This is reflected in the average CDF$_{total}$ of 0.52 ±0.02.

## 4. 3 La Campana

In situ-produced $^{10}$Be concentrations in La Campana range from (0.22 ±0.01) $10^5$ atoms/g$_{(qtz)}$ to (1.61 ±0.06) $10^5$ atoms/g$_{(qtz)}$ (Table S1). In La Campana the SPRs in the south-facing slope increase from top- to toe-slope with values of 42.1 ±2.6 to 290.5 ±24.0 t/(km$^2$ yr)) (Table S1). The SPR in the north-facing mid-slope position is slightly higher than the rate in the equivalent south-facing position (Fig. 2A; Table S1). Zirconium concentrations from rock to soil increase in La Campana

(Table S2). The chemical weathering rates in La Campana range from 20.2 ±2.6 t/(km$^2$ yr) to 210.5 ±34.3 t/(km$^2$ yr). About 60% of the chemical weathering occurs in the saprolite. The average total chemical weathering rate in La Campana of 95.0 ±44.7 t/(km$^2$ yr) is affected by a large standard error of the mean. This large error arises from the exceptionally high SPR of

sample LCPED30 likely related to the high topographic slope of 35°. The average $CDF_{total}$ is 0.49 ±0.07 indicating that half of the total denudation is due to chemical weathering (Table S4).

## 4. 4 Nahuelbuta

[10]Be concentrations in quartz range from (1.24 ±0.05) $10^5$ atoms/$g_{(qtz)}$ to (3.83 ±0.13) $10^5$ atoms/$g_{(qtz)}$ (Table S1). The SPRs from the south-facing slope in Nahuelbuta are within error, but increase slightly downslope from 57.7 ±3.5 to 68.6 ±4.2 t/(km$^2$ yr) (Table S1). The SPR in the mid-slope position of the north-facing slope is much lower than the rate from the mid-slope position in the south-facing slope (Fig. 2A; Table S1). The increase of zirconium from rock to soil is less pronounced in Nahuelbuta than in Santa Gracia or La Campana (Averaged $Zr_{soil}$/$Zr_{rock}$ ratio of 1.39 ±0.12; Tables S2 and S3). Chemical weathering rates indicate a wide range between 4.4 ±15.5 t/(km$^2$ yr) to 31.3 ±5.1 t/(km$^2$ yr). Physical erosion rates vary between 20.0 ±1.2 t/(km$^2$ yr) and 55.1 ±3.5 t/(km$^2$ yr). The average $CDF_{total}$ of 0.27 ±0.06 indicates that the physical erosion rate is higher than the chemical weathering rate (Tables S3 and S4). However, previous work by Oeser et al., (2018) highlights that a large variation in Zr concentrations within the regolith profiles of Nahuelbuta likely accounts for the high variability in CDF values.

In summary, SPRs generally increase from Pan de Azúcar to La Campana and are lower, or possibly equal, again in Nahuelbuta (Fig. 2A). From north to south, a similar picture is observed in the chemical weathering rates (Fig. 2B) as well as in the physical erosion rates (Fig. 2C). Therefore, the total denudation rate combining chemical weathering and physical erosion rates increases from Pan de Azúcar to La Campana and appears to decreases in Nahuelbuta to a slightly higher value than in Santa Gracia (Fig. 2D). The $CDF_{total}$ values and the contribution of $W_{total}$ to $D_{total}$ are almost zero in Pan de Azúcar. In Santa Gracia and La Campana they are about equal (~0.5), and then they are reduced in Nahuelbuta (~0.3; Table S4), although regolith heterogeneities in Zr concentrations in Nahuelbuta warrant caution when interpreting the CDF value here. Nevertheless, $W_{total}$ and $D_{total}$ of all samples from the four study areas correlate very well ($R^2$ = 0.96) with a slope of 0.50 (Fig. S3A).

## 5 Discussion

In the following sections, the soil production, chemical weathering, and physical erosion rates calculated for the Chilean study areas are discussed and compared to the global compilation. Whereas section 5.1 addresses SPRs, section 5.2 discusses findings and problems observed with chemical weathering and physical erosion rates.

### 5.1 Soil production rates

In this section SPRs from the Chilean Coastal Cordillera are compared to a global data compilation of SPRs from granitoid soil-mantled hillslopes in the light of MAP of the sample locations. The global data compilation from granitoid lithologies is

investigated for correlations of SPRs with topography, climate, and vegetation (leaf area index, LAI) metrics. Lastly, the SPRs are compared to model predicted variations in SPRs for different climate and vegetation settings.

### 5.1.1 Comparison of global SPR compilation based on MAP

The SPRs in the Chilean Coastal Cordillera increase from north to south from Pan de Azúcar to La Campana and are slightly lower, or possibly equal, again in Nahuelbuta (Fig. 2A). The maximum SPR of 290.5 ±24.0 t/(km$^2$ yr) is reached in La Campana with a MAP of 307 mm/yr. Comparison of MAP to the global compilation of SPRs from soil-mantled hillslopes on granitoid lithologies indicates that the maximum in SPRs (~900 t/(km$^2$ yr)) occurs at MAP values between ~500 to 600 mm/yr (Fig. 3A, Table S5). After this maximum, SPRs generally decrease with increasing MAP. This maximum in SPRs at MAP
values between ~500 to 600 mm/yr could be argued as the result of a sample bias because a significant amount of data in the global compilation (57 out of 303 samples; Table 2) are from the tectonically active San Gabriel Mountains (Heimsath et al., 2012). However, two sample sets (Table 2; Heimsath et al., 2005: Norton et al., 2012) from other tectonically active settings with high slopes (>30°; green symbols, Fig. 3A) and MAP of ~1000 to 1200 mm/yr support the observation that SPRs indeed decrease with increasing MAP. This trend is also observed for sample locations with lower slopes (e.g., 20 to 30°; yellow
symbols, Fig. 3A). Two samples from a mountain crest in the Jalisco Highlands (Riebe et al., 2004a) can be considered as outliers because the samples show high SPRs with reported low slopes (0 to 10°; red symbols, Fig. 3A). This is likely due to their location at a mountain crest with little or no soil. Attributing high slopes to these samples would again be in agreement with a reduction in SPR with increasing MAP.

In addition, the findings of the highest SPRs with steep slopes (>30°) are consistent with previous work (e.g., Binnie et al.,
2007) and agree well with the maximum SPRs for bare rock predicted from the 'maximum SPR model' of Norton et al. (2014; black line in Fig. 3A, see also Supplementary Material S1 for summary of model equations). This model predicts a linear increase in SPRs with increasing MAP. However, a purely linear relationship between SPRs and MAP is not evident for either the Chilean or global data sets for MAP > ~600 mm/yr, or for locations with slopes <30° (Fig. 3A). We note that although a limited number of study areas (n=4) are present in the Chilean data, the pattern observed of an increase and then decrease in
SPR with increasing MAP is present in both the Chilean and global datasets. This observation would be less clear if the datasets were plotted in log-log space as is commonly done (compare Figs 3A, 3C). Either this agreement between the Chilean and global datasets is pure coincidence and future studies will disprove it, or there is a systematic trend and underlying process that can explain this observation. If the latter is true then they could be consistent with existing model predictions, which we revisit in the Discussion section (see chapter 5.1.3).
Finally, we draw attention to a similar increase and then decrease in SPRs with increasing MAP that is observed for non-granitoid soil-mantled hillslopes (Fig. 3B). However, the maximum SPRs in these settings are not only higher than in granitoid lithologies but also reached at a higher MAP (~3,000 mm/yr;). This discrepancy between the trend in SPR with MAP for

granitoid versus non-granitoid catchments is perplexing, but is most easily explained by previous work that highlights the importance of lithology and rock strength variations when interpreting SPRs (e.g., Heimsath and Whipple, 2019).

The previously described increase and then decrease in SPR with increasing MAP is in contrast with the increase of SPRs predicted from the maximum SPRs for bare rock predicted with the model of Norton et al. (2014). The difference between observation and prediction could suggest that additional factors other than MAP influence the relationship. The observed decrease in SPR at MAP >600 mm/yr (Fig. 3A) could, for example, be attributed to other climate and vegetation effects on SPR and soil thickness (e.g., Langbein and Schumm, 1958; Amundson et al., 2015; Richardson et al., 2019; Mishra et al.,

2019; Starke et al., 2020). We underscore that the relationship between SPR and MAP identified here for Chile, and also present in the global compilation, are not a new result (e.g., Mishra et al., 2019), but have been obfuscated in previous studies by nature of the data being plotted in log-log (e.g., Fig. 3C) rather than linear plots (e.g., Fig. 3A and 3B and all other plots of this study). In the following, we investigate the non-linear effect of climate and vegetation on SPRs (for granitoid soil-mantled hillslopes only) with respect to different location specific parameters (e.g., MAT, LAI, slope, soil thickness).

**5.1.2 Correlation of SPRs with topography, climate, and vegetation parameters**

   As expected from the previous section, the SPRs of granitoid soil-mantled hillslopes and MAP are only weakly (inversely) correlated (R = -0.16; p < 0.05; Table S7). The best, but still weak, linear correlation of SPRs is achieved with local slopes (DEM GTOPO30; R = 0.37; p < 0.05; Table S7) indicating the potential importance of tectonic activity on hillslope angles and SPRs. Other parameters weakly correlating with SPRs are the mean altitude and MAT at the sample location (Table S7).

Remaining parameters investigated correlate even more weakly with SPRs than the previously mentioned slopes, mean altitude, and MAT correlations. In summary, based on the above linear correlation analysis, slope exerts the strongest control on SPR. Given this, in the following the non-linear relationship between SPRs and three parameters (MAP, MAT, and LAI) are investigated in more detail before we compare these relationships to different proposed models for SPR (Fig. 4). If we accept that the global data set is not biased due to sample availability, SPRs increase with increasing MAP and then decrease.

However, hillslopes with slopes >30° clearly document the highest SPRs (e.g., green symbols, Fig. 4) and SPRs increase with increasing slopes (Fig. S4). Therefore, we group SPRs into slope bins and apply polynomial fits to the different slope bins to qualitatively guide the readers eye to the structure of the data.

   We find that the maximum in the peak of the SPRs shifts to higher MAP values with decreasing slopes (Fig. 4A), but regardless of the slope the same general trend in the data is evident whereby SPRs initially increase, and then decrease towards higher

MAP. The relationship between SPR and MAT is less clear (Fig. 4B) aside from that SPRs generally decrease with decreasing slopes. Furthermore, there is a shift from a peak in SPRs at ~10ºC for slopes >30° (green line in Fig. 4B), to maximum values in SPRs occurring at low temperatures for slopes <20° (orange and red lines, Fig. 4B). In contrast, we find SPRs increase with increasing LAI to a maximum value of LAI = ~2.2 $m^2/m^2$ irrespective of slope (Fig. 4C). Furthermore, low SPRs can be found in landscapes with either thin soils and low MAP or thick soils and high MAP (Fig. S5). Whereas the former are landscapes

with low LAI, the latter are governed by high LAI. For example, in landscapes subjected to a dry climate (e.g., Pan de Azúcar

Chilean site), thin soils with little vegetation cause reduced SPR due to a combination of diminished abiotic and biotic weathering processes. Reduced SPRs are also observed in landscapes with a humid climate (e.g., the Nahuelbuta Chilean site) with thick soils, and abundant vegetation. As SPR depends on both chemical weathering and physical erosion a more detailed discussion is presented concerning patterns in chemical weathering and physical erosion rates (see Chapter 5.2).

### 5.1.3 Comparison of predicted climate and vegetation effects on SPRs

Previous observations of SPRs, climate, and vegetation have motivated different approaches to identify the functional relationship between these parameters. These studies stem from the empirically derived relationship proposed by Heimsath et al. (1997). Here we highlight three previous approaches (e.g., Fig. 5 and S6) aimed at quantifying the effects of MAP, MAT, and in some cases LAI, on SPRs (see Supplementary Material S1 for equations used). An overview of the approaches considered are as follows. First, Norton et al. (2014) developed an empirical soil production function to determine maximum SPRs based on MAP, MAT, and variable depth of soil cover (Fig. 5A and B). Second, Pelletier and Rasmussen (2009) used the "effective energy and mass transfer" (EEMT) approach of Rasmussen and Tabor (2007) to calculate SPRs (Fig. S6). Biotic controls are included in EEMT through the influence of MAP and MAT on biota, although we exclude this model from our discussion because it was based on river load, rather than hillslope-only analyses. Finally (third), Pelak et al. (2016) developed a production rate function assuming the abiotic production rate is constant, and the biotic effects are calculated based on present-day soil depth and vegetation biomass (Fig. 5C). The previous studies have differing assumptions and processes considered. As a result, they suggest markedly different functional relationships between MAP, MAT, vegetation, and SPRs, thereby highlighting uncertainty in our current knowledge and motivation for studies such as this one.

General trends in the controls on SPRs are, however, visible from observations. Maximum SPRs (under a constant average MAT of 14°C) are predicted to increase rapidly with increasing MAP to high values (Fig. 5A, black bold line). The maximum SPRs of Norton et al. (2014) explain the observed highest rates, but fail to capture the observed decrease in SPRs with increasing MAP (at MAP > ~600 mm/yr). Furthermore, the Norton et al. model predicts that if soil depths are thicker, then the slope of the line for maximum predicted SPRs decreases (Fig. 5A, black stippled lines).

However, the previously described approach of Norton et al. (2014) does not account for increases in soil depth that are observed with increasing precipitation and vegetation cover (e.g., Rasmussen and Tabor, 2007; Pelak et al., 2016). To explore the combined effects of variable soil depth with MAP (and hence vegetation cover) on SPRs we modified the approach of Norton et al. (2014) (see Supplementary Material S1) to account for increasing soil depth with increasing MAP (Fig. 5A; blue curve). Using this modified approach, the SPRs reach a maximum around 1,100 mm/yr MAP and then decrease with higher MAP. This modification to Norton et al., (2014) results in a predicted relationship more similar to global observations whereby SPRs initially increase, reach a maximum value, and then decrease as MAP increases. Although the blue curve in Fig. 5A does not provide a good fit to observations (in part because each sample location has a different MAT), we note that the general picture in the model predictions and data are more closely matched, thereby highlighting the potential importance of considering soil depth variations in this type of analysis. The relationship between MAT and SPR is less clear (Fig. 5B). Here,

the predicted maximum SPRs using the approach of Norton et al., (2014) are shown for different MAT (Fig. 5B), assuming no soil depth (black line) or stable soil depth (stippled lines). Comparison between the predicted and observed SPR shows a general disagreement between model predictions and observations using this approach.

The influences of MAP and MAT on SPR are often conceptualized as the main contributors to abiotic weathering. However, variations in MAP and MAT are controlling factors in vegetation type and amount, which potentially influence biotic weathering and SPRs through their demand for nutrients. Support for vegetation influencing SPRs comes from the work of Pelak et al. (2016) (Fig. 5C, see Supplementary Material S1 for equations). In their approach, SPRs are calculated as a function of biomass, which we represent with increasing LAI for comparison to our observations. This approach predicts an initial increase in SPRs with increasing LAI. The LAI value at which the maximum in SPRs occurs depends on the ratio of the vegetation growth rate (r) over the vegetation turnover time (m) (black curves Fig. 5C, see also Supplementary Material S1). The predicted increase then decrease in SPR shown in figure 5C is similar in the functional form to the globally observed increase then decrease in SPRs with LAI.

In summary, based on the previous considerations, we find that global variations in observed SPRs can be explained by variations in slopes, MAP, soil depth, and vegetation cover worldwide (Fig. 5). Our interpretation is based on diverse factors that influence vegetation cover and both the biotic and abiotic 'engines' contributing to soil production. In addition to MAP and soil depth, we find that SPRs vary with topographic slope (Fig. 4A and S4, see also Heimsath et al.; 1997) and MAT. Although available data do not allow a comparison of SPRs to rates of tectonic uplift, variations in tectonic uplift rates in the global compilation are indirectly represented by variations in hillslope angle. Given this, the relationships between SPR and MAP (e.g., Fig. 3) for diverse slopes suggests that regardless of the rate of tectonic uplift (likely manifested in different slopes) the trends in SPR documented here exist. More specifically, although high-slope settings produce the highest SPRs, changes in MAP and vegetation cause an increase and then decrease in SPRs as MAP or LAI increase, regardless of the hillslope angle (compare green and red lines, Fig. 4A, C).

## 5. 2 Chemical weathering, physical erosion, and total denudation rates

In this section chemical weathering and physical erosion rates, and CDFs for the Chilean Coastal Cordillera are discussed. The Chilean dataset is then compared to a global data compilation of soil-mantled hillslopes on granitoid lithologies (Fig. 6)

### 5.2.1 Data from the Chilean Coastal Cordillera

Before discussing the observations for chemical weathering, physical erosion and total denudation it is prudent to mention some of the complications associated with interpreting the data these processes are calculated from. For example, chemical weathering and physical erosion rates as well as CDFs are subject to uncertainties in Zr concentrations used for calculations. Zr concentrations in bedrock, saprolite, and soil can be highly heterogeneous due to lithologic variations in the parent material that affect the accuracy and reliability of calculated chemical weathering rates. For our Chilean sample locations, heterogeneity in Zr concentrations have been previously shown (Oeser et al., 2018) to be an issue for the southernmost study area (e.g.,

Nahuelbuta). Furthermore, bedrock outcrops available for sampling at neighboring locations within a study area may not be representative of the material the soil and saprolite formed from, as observed in the Santa Gracia Chilean study area (e.g., Fig. 9 and 10b in Oeser et al., 2018). Whereas this complication with Zr concentrations does not influence the calculation of our SPRs previously discussed, it does influence calculations of $W_{soil}$, $E_{soil}$, $W_{sap}$ and hence $D_{total}$. In the cases where the Zr

concentration of the average bedrock is lower than the Zr concentrations in soil and saprolite (Table S2), the calculated rates are reliable. In contrast, if the Zr concentration in the saprolite is lower than the concentration in the bedrock then $W_{sap}$, and hence $D_{total}$, needs to be interpreted with care because the calculated fraction of weathering in saprolite is negative and $D_{total}$ is underestimated. A too low concentration of Zr in saprolite is not consistent with the underlying assumption that soil production occurs from bedrock to saprolite and then soil (Table S2). Furthermore, a low concentration of Zr in the regolith may also

arise from aeolian and pedogenic input. For example, pedogenic gypsum in the soil is reported in the arid Pan de Azúcar study area (Bernhard et al., 2018) and suggests high aeolian input (e.g., see also Oeser and von Blanckenburg, 2020). However, this input only marginally affects the calculation of SPRs and $D_{total}$. Aeolian input could also be a concern in Nahuelbuta. Low bulk densities in regolith (0.8 ±0.1 g/cm$^3$) and high clay contents (>25 %; Bernhard et al., 2018) are suggestive of volcanic input. However, the chemical composition of the regolith and the pedogenic oxide content reveal no major volcanic influence

in Nahuelbuta (Oeser et al., 2018). Even though the calculations of chemical weathering rates are affected by large uncertainties some insights may be gained from the Chilean data. For example, in a plot of chemical weathering versus total denudation rate, supply- and a kinetically-limited weathering regime can be identified from these type of data (e.g., West et al., 2005; Ferrier et al., 2016). Whereas kinetically-limited weathering is thought to be dependent on climate and vegetation, a supply-limited weathering regime should not be influenced by climate or vegetation (West et al., 2005).

The data from the Chilean Coastal Cordillera reveal a strong linear relationship, with a slope of 0.5 in a plot of chemical weathering versus total denudation rate (Fig. S3A). This result indicates that weathering in the investigated settings is generally supply-limited and hence does not reflect a strong climatic nor vegetative influence (c.f., West et al., 2005). However, inspection of the CDFs from the four Chilean study areas opens up other perspectives. Average $CDF_{total}$ values of 0.05 ±0.08 to 0.52 ±0.02 are reported for the four study areas (Table S4). The low average $CDF_{total}$ value of Pan de Azúcar can be

explained by the low $W_{total}$ due to the absence of chemical weathering agents (e.g., carbonic acid produced from plant litter) and high pH values. Chemical weathering in this study area is kinetically-limited. In contrast, the low average $CDF_{total}$ value of 0.27 ±0.06 in Nahuelbuta cannot easily be explained by low $W_{total}$ caused by a lack of weathering agents. The effect of volcanic input on Zr concentrations and calculated $W_{soil}$ is also not applicable as no major volcanic influence is reported from Nahuelbuta. The low average $CDF_{total}$ could simply be attributed to the heterogeneity in lithology and Zr concentration encountered in Nahuelbuta (e.g., Oeser et al., 2018). However, alternative explanations could also lead to low $W_{total}$ and

$CDF_{total}$ values. First, the low values of $W_{total}$ in Nahuelbuta could be attributed to physical and/or chemical mobile behavior of the immobile elements (e.g., Taboada et al., 2006; Yoo et al., 2007), although inspection of Ti and Zr concentrations in the study areas suggest that for Zr, this is not likely (Oeser et al., 2018). Second, thick soils could lead to insolation of the saprolite

and reduce $W_{sap}$ (e.g., Burke et al., 2007). Related to this argument, are the findings of recent work (Oeser and von Blanckenburg, 2020) that suggests that the increase in biomass (supported by thicker soils and higher MAP) and associated higher nutrient demands in the south are accommodated by faster nutrient recycling, rather than nutrient acquisition from (difficult to access) underlying bedrock. If true, this interpretation could explain why the degree and rate of weathering along the transect is not strongly correlated with MAP because of changes in ecosystem nutrient acquisition vs. recycling processes. Third, the interplay of an increase in MAP and decrease in MAT governs the chemical weathering rates (e.g., Dixon et al., 2009) and co-variation in these parameters could complicate interpretations of $W_{sap}$. For example, from north to south in the Chilean study areas the increasing MAP could lead to increases in chemical weathering rates, while decreasing MAT could decrease the rates, making a straightforward interpretation of the data more difficult. Finally (fourth), weathering rates in Nahuelbuta could also be influenced by differences in microbial activity (e.g., Buss et al., 2005; Eilers et al., 2012) relative to the other study areas. MAP, MAT, and pH all affect microbial abundance (e.g., Fierer and Jackson, 2006; Bahram et al., 2018; Tan et al., 2020). For example, previous work has documented that microbial abundance can increase and then decrease along a tropical elevation-climate gradient where MAP increases and MAT decreases with increasing elevation (Peay et al., 2017). A comparable increase in MAP and decrease in MAT is observed in the Chilean Coastal Cordillera as in the previous tropical elevation-climate gradient, which may explain the decreasing pH values from north to south and the lower bacterial abundance in Nahuelbuta than La Campana (Bernhard et al., 2018; Oeser et al., 2018). To investigate if the low average $CDF_{total}$ values reported from Nahuelbuta are meaningful or just a result of the observed heterogeneity in lithology and/or Zr concentrations, we proceed to compare data form this study with a global data compilation from granitoid soil-mantled hillslopes.

### 5.2.2 Comparison to global data compilation

Similar to the Chilean Coastal Cordillera, the global data compilation also shows an increase and then decrease in the chemical weathering, physical erosion, and total denudation rates with increasing MAP (Fig. 6). These observations are caused by the interaction of climate, tectonics, and the biosphere at the Earth's surface. However, the interactions are difficult to disentangle due to a variety of factors that are either unknown, or known and difficult to quantify with existing data. For example, previous work by Oeser et al. (2018) documented mineralogical and geochemical differences between lithologies of the Chilean study areas. While we have restricted our analysis (as much as possible) to "granitoid lithologies" there are still compositional differences between the different types of granitoids present. The mineralogical differences between different granitoid rocks could influence the rates of chemical weathering. This concern also applies to our global compilation of granitoid sample locations. Although we filtered study areas shown by rock type, significant work remains to be done in quantifying chemical weathering rates within different types of granite (e.g., Lebedeva and Brantley., 2020). Furthermore, the Chilean and global datasets contain geographic variations in rates of rock uplift. Our approach here was to identify if chemical weathering and physical erosion rates correlate with topographic slope, which is often considered a proxy for rock uplift rate. This connection is made as high slopes commonly occur in areas with high rates of rock uplift and lower slopes occur in more tectonically quiescent settings. While this approach is often used for a qualitative comparison, it does not provide a direct quantitative

comparison between rates of tectonic activity and chemical weathering and physical erosion processes. To address this concern, a global compilation of rock uplift rates, and other potentially related factors such as fracture density, would be needed for the correlation analysis. Unfortunately, these data sets do not exist. In addition, this study evaluated not only how present-day climatology (precipitation and temperature) compare to observed SPR and weathering rates, but also how these results would differ based on available paleo-precipitation and temperature estimates for the Middle Holocene and LGM (Table S5). This approach is useful for evaluating if a legacy of past climate conditions correlates with observed weathering and erosion processes (which it does not in our case). A similar conclusion was reached for the western Andean margin (Precordillera) of Peru and Chile in work by Starke et al. (2020). However, our approach does not address the full range of potentially relevant paleo-conditions. For example, while changes in LGM or Mid Holocene precipitation and temperatures exist, their effect on paleo-solute fluxes, or temperature dependent paleo-microbial abundances remain unquantified as to their significance (if any). These paleo-effects on weathering and erosion rates warrant future investigation.

Finally, there are also several methodological aspects of this study that need to be considered. In all the Chilean and global data compilations presented a replicate analysis at each sample location is not available. For the Chilean data (and often times in many of the global studies referenced) the common practice is to collect an amalgamated sample across an exposed horizon (often times ~1 m wide, the width of the pedon) for analysis. Due to the considerable effort involved in excavating pedons, rarely is a replicate analysis at all depths conducted in these types of geochemical studies. The implication is that analysis of results with replicate sample locations reduces the types of statistical analyses that can be done to understand how representative the sample location is. In this study we compared observed soil production, chemical weathering, and physical erosion rates to globally derived datasets for topography, and climate (Table S7 and S9). This approach ensures that all data are treated consistently, but does not evaluate how representative these global compilations are to more regional data sets. For example, the CHELSA climate data set used here is a downscaled (to ~900 m) product derived from the ERA-interim reanalysis data. The ERA-interim data is widely used in the climate sciences and is derived from ground, ocean, and satellite meteorological observations. However, more regional reanalysis data products exist for some of the study areas covered in our global compilation that might capture orographic effects in precipitation better than a 900 m resolution global product. However, such regional downscaled products are not available in many places (globally) and there are diverse modeling, statistical, and machine learning techniques applied for downscaling climate data. Furthermore, not all regional climate data is peer-reviewed. As a result, a mixing and matching of different regional climate products (produced via different techniques) versus using global data products (produced with a consistent downscaling approach everywhere) would lead to its own, technique based, uncertainties and biases. Given these types of complications, we have restricted our analysis to global data products produced with a consistent set of techniques, but acknowledge that future global data products conducted at higher resolution with longer observational records constraining them may provide an improved analysis to build upon the results presented here.

Keeping all the previous limitations in mind, it is again astonishing that the global dataset shows a similar picture as the Chilean dataset when plotted versus MAP. A similarity between the Chilean and global data is therefore not only observed for the

SPR data presented, but also for the chemical weathering and physical erosion rates as well as the CDF. If this similarity in observations is not coincidence, then the CDF values may be interpreted in the following way. Values of CDF are generally around 0.4 to 0.6 in the global data compilation (Fig. 6 and S3B). However, values below and above 0.4 to 0.6 are also observed (Fig. 6). CDF values above 0.6 are rare. For example, CDF values corrected for the chemical erosion fraction after
565 Riebe and Granger (2013) from Puerto Rico and McNabb Track in New Zealand are higher than 0.6 (Table S8). More often, CDF values are lower than 0.4 and may even be close to zero indicating that chemical weathering is not operative or occurs only at low rates (e.g., Riebe et al., 2004a; Norton and von Blanckenburg, 2010; Ferrier et al., 2012). CDF values below 0.4 are reported from dry and hot places with low vegetation where water as a weathering agent is rare (e.g., Pan de Azúcar, this study). Other settings with low CDF values were previously encountered in cold regions with frost cracking and/or snow (e.g.,
Idaho Batholith, Ferrier et al., 2012; Switzerland, Norton and von Blanckenburg, 2010). Chemical weathering in cold regions might be low due to low temperatures, which in turn enable frost cracking and snow to increase physical erosion rates and diminish CDFs (Table S8). However, not only do climate and vegetation influence values of CDFs, but so does topography. For example, low CDFs are reported from Point Reyes in California by Burke et al. (2007). This study area is underlain by a granodioritic lithology and located in a mediterranean climate with a reported 800 mm/yr of precipitation (Heimsath et al.,
2005). The vegetation cover is grassland, shrubs, and trees, and hosts pocket gophers. CIA values of 59 for bedrock and 63 to 91 for soils (average 81) are reported. CDF values range from 0.06 to 0.75 with an average of 0.26. Average CDF values of thin soils (<60 cm) in divergent slope positions are 0.51, whereas thick soils (>60 cm) in a divergent position and soils in convergent positions indicate an average value of 0.13. Burke et al. (2007) concluded that thick soils act as a buffer against intense chemical weathering and inhibit high chemical weathering rates. In summary, the decrease of CDFs with increasing
MAP could be explained by reduced chemical weathering of saprolite due to buffering by thick soils (e.g., Burke et al., 2007). The protective nature of thick soils may distance vegetation from "fresh" nutrients in bedrock thereby leading to nutrient recycling (e.g., Jobbágy and Jackson, 2001; Oeser and von Blanckenburg, 2020; Koester et al., 2020) rather than nutrient acquisition from unweathered bedrock. This finding is consistent with previous work that has identified a decrease in chemical weathering rates with increased soil thickness as well as a 'speed limit' to chemical weathering rates (e.g., Dixon and von
Blanckenburg, 2012). In essence, dense vegetation could be functioning as a 'biotic break' on soil production and weathering. If true, then these observations suggest that biotic and abiotic weathering processes need to be considered in tandem when interpreting SPR, and chemical weathering and physical erosion rates.

## 6 Summary and Conclusion

Observed soil production, chemical weathering, and physical erosion rates are variable along the climate and vegetation
gradients in the Chilean Coastal Cordillera. Whereas vegetation cover and mean annual precipitation increase from north to south, mean annual temperatures decreases. SPRs increase from Pan de Azúcar to La Campana and decrease slightly, or possible stay the same, in Nahuelbuta. Chemical weathering rates increase from zero in the north to a maximum of 211 t/(km$^2$

yr) in La Campana. Physical erosion rates are low in Pan de Azúcar (~10 t/(km$^2$ yr)), increase towards the south and are highest in La Campana. Combined total chemical weathering and physical erosion rates indicate that the total denudation rates are lowest in Pan de Azúcar and highest in La Campana. The contribution of chemical weathering to the total denudation rate is negligible in the north, and increases to ~50% towards the south. The observations made in the Chilean Coastal Cordillera and our comparison to global data lead to the following conclusions for the hypotheses stated in section 1.0:

1)    Calculated soil production and chemical weathering rates do not appear in either dataset to increase monotonically with increasing MAP, except for at low (<~600 mm/yr) MAP. This result suggests a weak climate control on SPR and chemical weathering rates (i.e., supply-limited, c.f., West et al., 2005). The decrease in SPRs at higher precipitation and vegetation cover is likely due to thicker and more nutrient depleted soils in settings with higher vegetation/LAI. The thick and clay-rich soils may not only provide a protective cover for intense weathering of the saprolite, but also cause vegetation to switch to a nutrient recycling mode of near surface material (e.g., Jobbágy and Jackson, 2001; Oeser and von Blanckenburg, 2020) rather than harvesting fresh nutrients from deeply buried bedrock that would facilitate continued soil production. If true, this mechanism suggests a biotic influence on critical zone processes in temperate settings. Thus, vegetation cover could be providing the 'biotic break' in these settings and responsible for a speed limit and deceleration in observed rates. Another explanation could be the latitudinal variations in microbial abundance due to climate and soil pH conditions influencing biotically driven chemical weathering rates.

2)    The contribution of chemical weathering to total denudation rates appears to increase and then decrease, or stagnate with increasing MAP. This is apparent in both the Chilean and global data sets. In dry areas, the low contribution in chemical weathering to total denudation is due to the lack of a weathering agent. However, in wet areas the low CDF values cannot be easily explained and several (currently untestable) explanations could hold true for the Chilean dataset. A key factor influencing a more robust interpretation of CDF values lies in heterogeneities in the Zr concentrations that warrant further investigation. However, despite these uncertainties in the Chilean dataset, variations in chemical weathering and total denudation rates observed along the four Chilean study areas are also manifested at the larger global scale hinting that similar processes are in play across scales.

To conclude, we find soil production, chemical weathering, and physical erosion rates change with increasing MAP and LAI non-linearly. Whereas increasing MAP has the potential to increase chemical weathering and physical erosion rates, vegetation decreases physical erosion rates and to a certain extent also chemical weathering rates in temperate climate zones.

**Data availability**

Data used in this study are accessible in tables S1 to S9 as a data supplementary in Schaller and Ehlers (202x).
These data are freely available under the Creative Commons Attribution 4.0 International (CC BY 4.0) open access license at GFZ data services.

When using the data please cite: Data supplement to: Comparison of soil production, chemical weathering, and physical erosion
rate along a climate and ecological gradient (Chile) to global observations. GFZ data services. (NOTE from the auhors to
reviewers and editor: We provide the data as part of this submission for your viewing, but if this manuscript is accepted, the
data files will also be assigned a doi number through GFZ data services and we will include a link in this manuscript to assure
data longevity).

**Author contribution**

Both authors (MS, TAE) contributed equally to the planning, execution, and manuscript preparation.

**Competing interest**

The authors declare that they have no conflict of interest.

**Acknowledgement**

We would like to thank the Chilean National Forestry Corporation (CONAF) for providing access to the sample locations and
635 on-site support of our research, and the broader EarthShape research consortium for stimulating discussions. Finally, we thank
two anonymous reviewers and the handing editor for their insightful comments which improved the manuscript.

**Financial support**

This work was funded by the German Science Foundation (DFG) priority research program SPP-1803 "EarthShape: Earth
Surface Shaping by Biota" (grant SCHA 1690/3-1 to MS and EH329/14-2 to TAE).

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

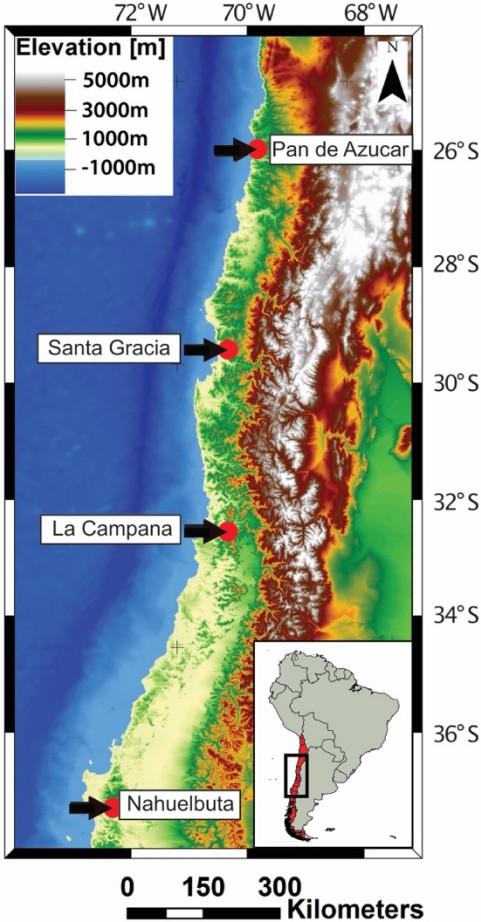

**Figure 1:** Digital elevation model (Data source: GTOPO30) with the locations of the four study areas (red circles) in the Chilean Coastal Cordillera. The climate from north to south over ~1200 km changes from arid to temperate humid. See Supplementary Material figure S1 photographs of each area and detailed satellite images of sample locations.

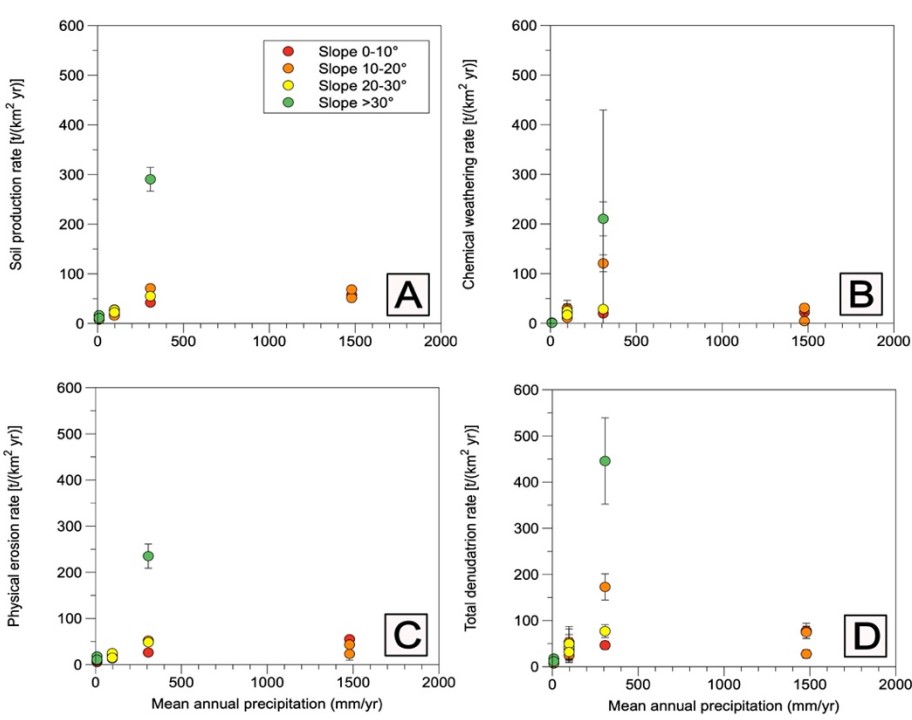

**Figure 2: Cosmogenic nuclide-derived rates from pedon locations for four study areas in the climate and vegetation gradient of the Chilean Coastal Cordillera (From north to south and with increasing mean annual precipitation: Pan de Azúcar, Santa Gracia, La Campana, and Nahuelbuta): A) Soil production rates, B) chemical weathering rates, C) physical erosion rates, and D) total denudation rates versus mean annual precipitation**

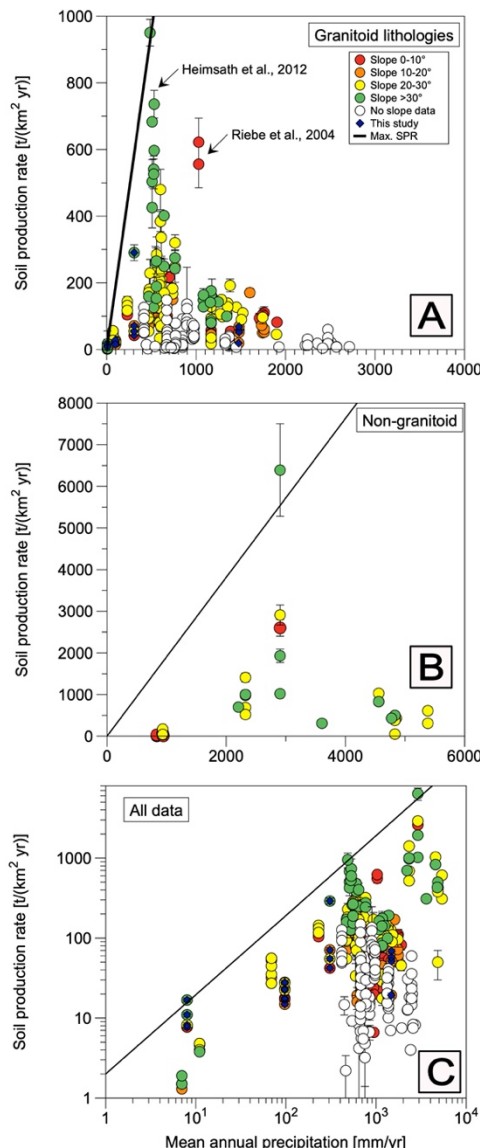

**Figure 3: Soil production rates SPRs versus mean annual precipitation: A) SPRs from granitoid lithologies such as samples from this study (blue diamonds). The data are in bins for different slopes. Note the linear axes as well as the black line for the maximum production rate after Norton et al. (2014) at a temperature of 14ºC. B) SPRs from some non-granitoid lithologies. Note the increased linear axis in comparison to Fig. 3A. C) Compilation of all SPRs plotted with logarithmic axis.**

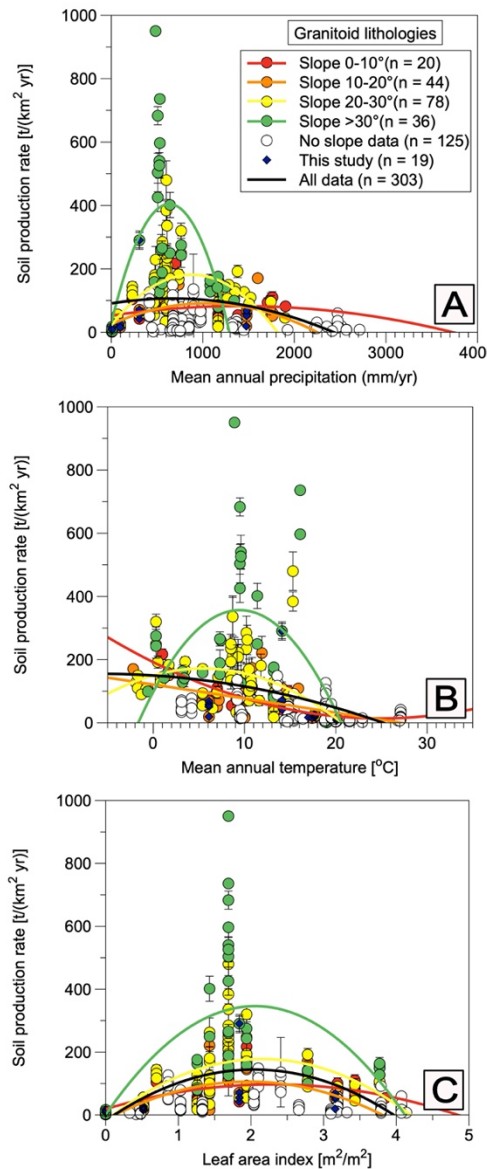

**Figure 4: Soil production rates from this study (blue diamonds) and other studies in hillslopes with granit lithologies versus: A) Mean annual precipitation, B) Mean annual temperature, and C) Leaf area index. Data are separated in slope bins and plotted with a binomial fit. Curves shown in subplot were calculated with polynomial regressions through the data to help visualize the trend in observations for locations with different hillslope angles.**


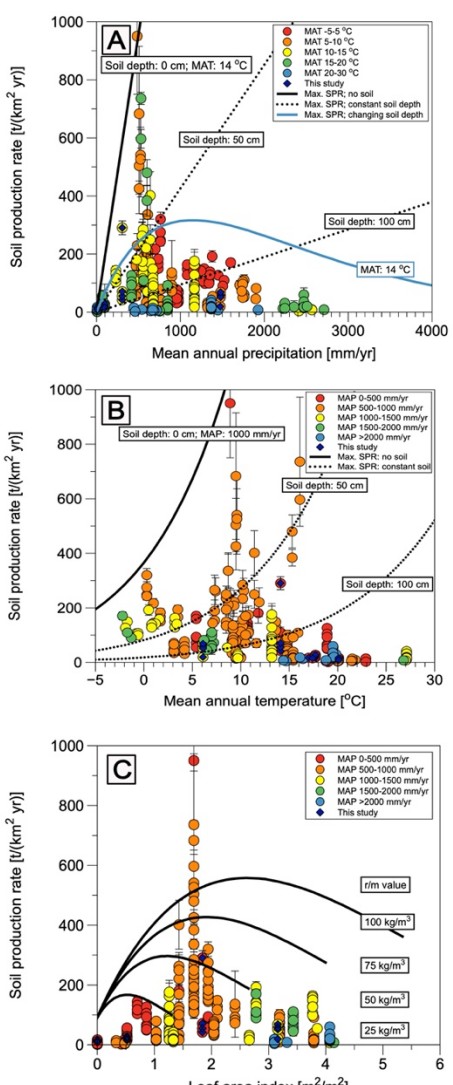

**Figure 5: Soil production rates versus: A) Mean annual precipitation. The black bold and stippled lines indicate maximum SPRs for different soil depths (after Norton et al., 2014). The blue line is the maximum SPR with increasing soil depth with increasing mean annual precipitation. Data are sorted in mean annual temperature bins. B) Mean annual temperature. Black and stippled lines are maximum SPRs for different soil depths. Data sorted in mean annual precipitation bins. C) Leaf area index. Black lines are maximum SPRs for different biomasses (r/m value; after Pelak et al., 2016). Data sorted in mean annual precipitation bins.**

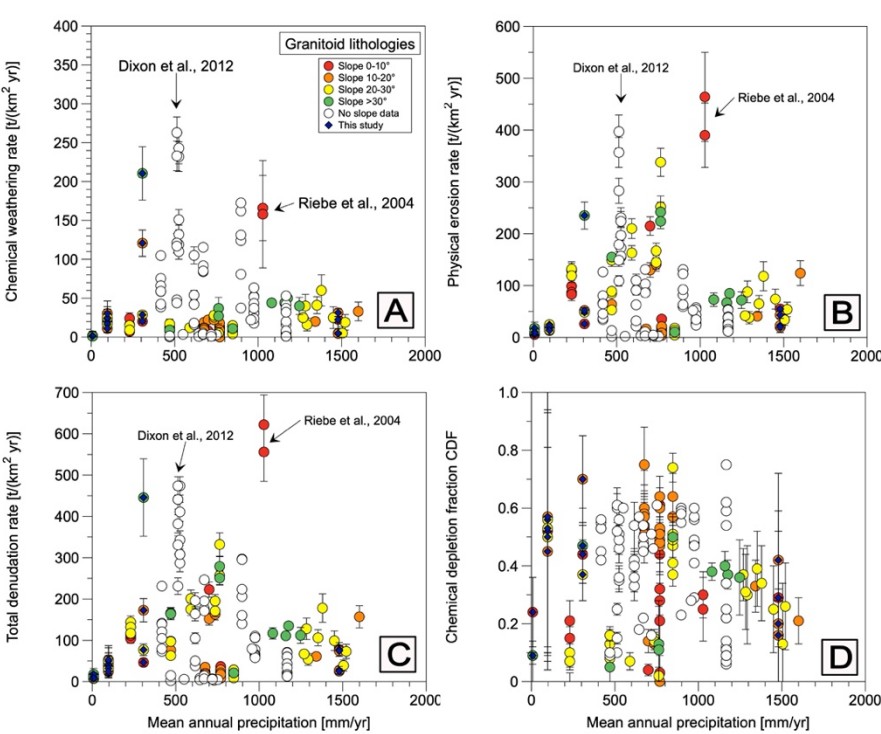

**Figure 6: A) Chemical weathering rates, B) Physical erosion rates, C) Denudation rate, and D) Chemical depletion fraction CDF versus mean annual precipitation. Rates for the four study areas (blue diamonds) and comparison rates from different study areas situated in granitoid soil-mantled hillslopes (slope sorted).**


Table 1: Location for pedons in the investigated four study areas in the Chilean Coastal Cordillera

| Soil profile | Location °N | °E | Altitude m | MAP mm/yr | MAT °C | Veg. cover % | LAI m$^2$/m$^2$ | Aspect ° | Slope ° | Soil depth cm | Ave. regolith density g/cm$^3$ |
|---|---|---|---|---|---|---|---|---|---|---|---|
| **Pan de Azucar** | | | | | | | | | | | |
| AZPED60 | -26.11012 | -70.5492 | 343 | 8 | 20.1 | 2 | 0 | 60 | 5 | 22 | 1.26 |
| AZPED50 | -26.11027 | -70.54922 | 333 | | | | | 0 | 40 | 20 | 1.47 |
| AZPED40 | -26.1102 | -70.5492 | 326 | | | | | 0 | 33 | 25 | 1.28 |
| AZPED21 | -26.1094 | -70.54907 | 342 | | | | | 180 | 25 | 20 | 1.31 |
| **Santa Gracia** | | | | | | | | | | | |
| SGPED10 | -29.75633 | -71.16626 | 727 | 97 | 17.7 | 31 | 0.52 | 300 | 2 | 5 | |
| SGPED20 | -29.75636 | -71.16721 | 718 | | | | | 240 | 2 | 30 | 1.53 |
| SGPED30 | -29.75702 | -71.16650 | 705 | | | | | 0 | 25 | 35 | |
| SGPED40 | -29.75738 | -71.16635 | 682 | | | | | 0 | 25 | 50 | 1.48 |
| SGPED50 | -29.75794 | -71.16618 | 652 | | | | | 0 | 25 | 40 | |
| SGPED60 | -29.75826 | -71.16615 | 638 | | | | | 0 | 20 | 55 | 1.48 |
| SGPED70 | -29.76120 | -71.16559 | 690 | | | | | 180 | 15 | 35 | 1.55 |
| **La Campana** | | | | | | | | | | | |
| LCPED10 | -32.95581 | -71.06332 | 734 | 307 | 14.1 | 84 | 1.84 | 60 | 7 | 45 | 1.37 |
| LCPED20 | -32.95588 | -71.06355 | 718 | | | | | 0 | 23 | 60 | 1.21 |
| LCPED30 | -32.95615 | -71.06380 | 708 | | | | | 60 | 35 | 55 | 1.22 |
| LCPED40 | -32.95720 | -71.06425 | 724 | | | | | 120 | 12 | 35 | 1.47 |
| **Nahuelbuta** | | | | | | | | | | | |
| NAPED10 | -37.80735 | -73.01285 | 1248 | 1479 | 6.1 | 95 | 3.16 | 60 | 5 | 70 | 0.94 |
| NAPED20 | -37.80770 | -73.01357 | 1239 | | | | | 60 | 15 | 70 | 0.91 |
| NAPED30 | -37.80838 | -73.01345 | 1228 | | | | | 0 | 20 | 90 | 0.88 |
| NAPED40 | -37.80904 | -73.01380 | 1200 | | | | | 180 | 13 | 60 | 1.05 |

Table 2: Overview of published rates from granitoid soil-mantled hillslopes

| Refernce | Study area | Data amount | Latitude | Approximate Longitude | Altitude |
|---|---|---|---|---|---|
| | | | N | E | m |
| **Soil production rates** | | | | | |
| This study | Coastal Cordillera, Chile | 4 | -26.11 | -70.55 | 336 |
| | Coastal Cordillera, Chile | 7 | -29.76 | -71.17 | 666 |
| | Coastal Cordillera, Chile | 4 | -32.96 | -71.06 | 721 |
| | Coastal Cordillera, Chile | 4 | -37.81 | -73.01 | 1229 |
| Owen et al., 2011 | Coastal Cordillera, Chile | 5 | -24.13 | -69.99 | 1170 |
| | Coastal Cordillera, Chile | 5 | -26.28 | -70.49 | 687 |
| | Coastal Cordillera, Chile | 8 | -29.77 | -71.08 | 377 |
| Heimsath et al., 2005 | Point Reyes, USA | 15 | 38.08 | -122.88 | 165 |
| Heimsath et al., 2000 | Bega Valley, Australia | 15 | -36.62 | 149.50 | 391 |
| Heimsath et al., 2009 | Arnhem Land, Australia | 13 | -12.47 | -133.29 | 140 |
| Heimsath et al., 2006 | Bega Valley, Australia | 18 | -36.62 | 149.50 | 807 |
| | Bega Valley, Australia | 11 | -36.78 | 149.72 | 220 |
| Heimsath et al., 2001 | Frogs Hollow, Australia | 11 | -36.00 | 149.00 | 921 |
| Dixon et al., 2009 | Sierra Nevada, USA | 7 | 36.95 | -119.63 | 216 |
| | Sierra Nevada, USA | 4 | 37.04 | -119.25 | 1186 |
| | Sierra Nevada, USA | 6 | 37.07 | -119.21 | 1952 |
| | Sierra Nevada, USA | 5 | 37.28 | -119.11 | 2681 |
| | Sierra Nevada, USA | 9 | 37.28 | -119.09 | 2991 |
| von Blanckenburg et al., 2004 | Central Highlands, Sri Lanka | 18 | 7.22 | 80.79 | 1504 |
| Riebe et al., 2004 | Jalisco Highlands, Mexico | 2 | 20.35 | -105.50 | 750 |
| | Santa Rosa Mountains, USA | 6 | 41.33 | -117.63 | 2377 |
| Byun et al., 2015 | Daegwanryeong Plateau, South Kor | 11 | 37.76 | 128.71 | 1096 |
| Riggings et al 2011 | Bodmin Moor, United Kingdom | 6 | 50.55 | -4.60 | 377 |
| Heimsath et al., 2012 | San Cabriel Mountains, USA | 57 | 34.33 | -117.94 | 1826 |
| Norton et al., 2010 | Swiss Alps, Switzerland | 18 | 46.49 | 8.24 | 2321 |
| Ferrier et al., 2012 | Idaho Batholith, USA | 11 | 45.16 | -115.30 | 1621 |
| | Idaho Batholith, USA | 6 | 45.07 | -115.30 | 1681 |
| Heimsath et al., 2020 | Kruger National Park, South Africa | 2 | -23.04 | 31.25 | 323 |
| | Kruger National Park, South Africa | 3 | -25.03 | 31.50 | 348 |
| | Kruger National Park, South Africa | 1 | -24.41 | 31.54 | 383 |
| | Kruger National Park, South Africa | 4 | -25.02 | 31.50 | 356 |
| | Kruger National Park, South Africa | 7 | -25.20 | 31.28 | 569 |
| **Chemical weathering, py\hysical erosion, and total denudation rates** | | | | | |
| This study | Coastal Cordillera, Chile | 4 | -26.11 | -70.55 | 336 |
| | Coastal Cordillera, Chile | 6 | -29.76 | -71.17 | 666 |
| | Coastal Cordillera, Chile | 4 | -32.96 | -71.06 | 721 |
| | Coastal Cordillera, Chile | 4 | -37.81 | -73.01 | 1229 |
| Riebe et al., 2004 | Jalisco Highlands, Mexico | 2 | 20.35 | -105.50 | 750 |
| | Santa Rosa Mountains, USA | 6 | 41.33 | -117.63 | 2377 |
| Burke et al., 2007 | Point Reyes, USA | 8 | 38.08 | -122.88 | 165 |
| Burke et al., 2009 | Bega Valley, Australia | 8 | -36.78 | 149.72 | 220 |
| | Bega Valley, Australia | 10 | -36.62 | 149.50 | 807 |
| | Frogs Hollow, Australia | 16 | -36.00 | 149.00 | 921 |
| Dixon et al., 2009 | Sierra Nevada, USA | 7 | 36.95 | -119.63 | 216 |
| | Sierra Nevada, USA | 4 | 37.04 | -119.25 | 1186 |
| | Sierra Nevada, USA | 5 | 37.07 | -119.21 | 1952 |
| | Sierra Nevada, USA | 1 | 37.28 | -119.11 | 2681 |
| | Sierra Nevada, USA | 9 | 37.28 | -119.09 | 2991 |
| Norton and von Blanckebburg, 2010 | Swiss Alps, Switzerland | 17 | 46.50 | 10.97 | 2361 |
| Ferrier et al., 2012 | Idaho Batholith, USA | 11 | 45.16 | -115.30 | 1621 |
| | Idaho Batholith, USA | 6 | 45.07 | -115.30 | 1681 |
| Dixon et al., 2012 | San Cabriel Mountains, USA | 17 | 34.36 | -118.01 | 1694 |
| Heimsath et al., 2020 | Kruger National Park, South Africa | 2 | -23.03 | 31.27 | 323 |
| | Kruger National Park, South Africa | 6 | -25.02 | 31.50 | 352 |
| | Kruger National Park, South Africa | 4 | -25.21 | 31.28 | 560 |