# Peer review of "Comparison of soil production, chemical weathering, and physical erosion rates along a climate and ecological gradient (Chile) to global observations"

_Earth Surface Dynamics, 2021_

## Author Response (AR2)

**General comment to the handling editor**

We thank the handling editor for his insightful comments which are addressed below. We are also very thankful for your patience in allowing an extension in submitting the revised version due health complications that have arisen. In the revised manuscript we have made extensive (additional) changes to address not only your comments below, but also those of the external reviewers. In the track changes version of the manuscript, we show the differences between the original submission and our revisions. Overall, we found your and the reviewer's comments useful and we extensively revised the text, some figures, and added additional figures /tables described below.

The comments of the handling editor are in bold, the replies in italics, and changed text in normal font.

Best wishes,
Mirjam Schaller (corresponding author)

**Comments to the Author:**
**Dear Authors,**

**Thank you for replying to the referee comments. Based on my independent assessment, and my reading of the response and the revisions proposed so far, I call for major revisions. At that point I plan to consult the reviewers again.**

**Please consider these comments which summarise remaining issues, and provide a reply along with revisions:**

**1) Both reviewers note concern in over-reaching from the Chilean study sites to the compilation provided. As reviewer 1 notes - there is the "real danger of overinterpretation". I found the response and revisions not to be satisfactory on this point. More work is needed. This is relevant for both the Chilean dataset, and the use of the global complication. The authors added a caveat section, but this needs to distributed throughout the manuscript, and a more careful tone adopted from the title, abstract, introduction through the study design.**

*We have revised the manuscript and tried to adopt a more careful tone in throughout the entire manuscript. In order to do so the following changes were performed:*
*- Reduce paper to the presentation of new data from the Chilean Coastal Cordillera and comparison to global data compilation of granitoid soil-mantled hillslopes.*
*- Distribution of the caveats section into individual chapters where a concern is encountered. This was done throughout the methods, results, and discussion sections.*
*- Discuss more extensively the effect of possible inhomogeneities in the source rocks for chemical weathering rates (Chapter 5.2.1).*
*- Mention Zr mobility as a possible but un-likely possibility (Chapter 5.2.1).*
*- Reduce and tone down discussion of CDF.*

**The title suggests global patterns, yet the abstract focuses on the Chilean study.**

*The title is changed to: "Comparison of soil production, chemical weathering, and physical erosion rates along a climate and ecological gradient (Chile) to global observations" which accurately describes the content of the manuscript.*
*The abstract has been significantly revised and mentions the comparison of the Chilean to global data. We hope that these changes improve the manuscript.*

**In the abstract, results and discussions, the Chilean data are described in misleading ways, referring to "trends" etc., when there are only 4 sites (4 distinct MAP) being described. For instance, the language between lines 16-18 simply does not reflect what is shown in Figure 2 (similar text is found in the introduction, results, discussion). I would describe the data in a different way - the lowest MAP (which is very low) is where SPR is lowest, but then there is simply not enough data to establish any distinct patterns of soil production rate and MAP.**

*The results from the four sites with distinct MAP are no longer described to show a trend. We have replaced with the word 'trend' throughout the data with alternative wording. Instead, we talk in the entire manuscript about the range of the data and desribe the rates as highest or lowest.*

**So, there is work to be done throughout the manuscript to adopt a more cautious tone that reflects the patterns in the data (or lack thereof) which are plain to see.**

*We hope that the changes and corrections as requested above are performed to satisfy the handling editor. However, we'd also like to bring to the editor's attention that many other studies of this type plot the data in log-log space which makes it easier to describe relationships in the data, but misses the signal we intend to draw attention to in this study (i.e. a consistent observed Chilean and Global decrease in soil production rates (and other parameters) at high precipitation rates where dense vegetation cover is present.*

**2) Any sampling bias in the global compilation: Soil production rates have generally been measured in places where soils are thick - and this biases the datasets to places where weathering may be limited by supply. This is linked to the Dixon, Heimsath and Larsen papers (and the speed limit or not for SPR). When the authors use those datasets here, we loose appreciation that the sample sets were collected with those themes in mind. For instance, we don't have a higher MAP equivalent of Heimsath in a pure felsic bedrock - so we don't know what that might look like.**

*The handling editor's concern of sampling bias in the global data set is justified. We addressed this concern in two ways.*

*First, an extensive paragraph was added in section 3.3 (methods) to address assumptions and potential limitations of the global compilation. In doing this, we also provide additional (and admittedly needed) data on the criteria used in our global compilation. Your concern raised above about a sampling bias in global SPR studies is addressed in this section.*

*Second, (and here we slightly disagree with the editor) It is true that the sample set of SPRs from the San Gabriel Mountains (Heimsath et al., 2012) with high SPRs and high slopes is dominant (e.g., see new Table 2 and Fig. 4a). However, there is another cluster of data points at ~1200 mm/yr MAP with as steep slopes but not as high SPRs as in the San Gabriel Mountains. The data stem mainly from the Swiss alps (Norton and von Blanckenburg, 2010) as well as from Point Reyes in California (Heimsath et al., 2005). This cluster at double the MAP as in the San Gabriel Mountains but lower SPRs seems to support the observation also made in the Chilean Coastal Cordillera that SPRs decrease with increasing MAP. Either this match is pure coincidence and future studies will disprove the observation or there is an underlying process that can explain this observation. As the latter could be true, we further explore if existing models in the literature are consistent with this observation, which in fact some of them are. Thus, we provide a range of observations that show variations with precipitation and vegetation LAI, and also process based models that explain the functional form of these observations.*

*In summary, in order to address the concern of the handling editor, we discuss our findings in the text, report the number of plotted samples in Fig. 4a, and add a new Table 2 containing information about sample amounts and origin. Of course, the concern still exists that more data might invalidate the observation, but with the available data and the model consideration the described observations appear to be real. We find it rather striking that the results observed in the Chilean data set, when compared to global compilations using fairly strict criteria (see new text), produce similar results. The patterns in the observations presented here are also present in previous global analyses, but were obscured by authors plotting results in log-log.*

*We hope you find these changes and line of argumentation acceptable.*

**Except that we have a hint from Larsen et al., 2014, which the reviewer #1 points out has a higher MAP than is used here, and would suggest that high MAP and high erosion can drive high MAP.**

*Assuming that this sentence should be "…that the high MAP and high erosion can drive high SPRs" we agree that: 1) the MAP reported here based on Karger et al. (2017) is smaller than that reported from the local weather station data used in the Larsen et al. study; and 2) the SPRs from Larsen et al. (2014) are higher than the rates in the data set for granitoid lithologies. The former (1) does influence the findings in such a way that the SPRs are just shifted to higher MAP, but the observation of lower SPRs with higher MAP remains the same. Thus, this effect would not influence the level at which we interpret the data. Furthermore, we have added text at the start of section 3.3 (Global compilation methods) to also state (and justify) why we use global climate (reanalysis) data compilations. It is well known in climatological communities that station meteorological data differ from reanalysis data. This is due to the comparison being made between a point measurement (met. Station) and the larger spatial scale average behavior represented in reanalysis data. If many more (tens or hundreds) of meteorologic*

*stations were available within the spatial area considered within a reanalysis data set then two should agree more closely.  One of the motivations for different reanalysis data processing techniques (which use point and satellite data as inputs) is to provide a physically meaningful interpolation of point measurements across larger spatial scales.*

*The latter (2) is also not surprising as lithology is recognized to influence SPRs (e.g., Heimsath et al., 1997). Therefore, the compiled SPRs from granitoid soil-mantled hillslopes (interpreted in this study) are expected to be different than the SPRs of Larsen et al. (2014) from soil-mantled hillslope developed in schist. We make this point several times in the manuscript and show it in Figure 3B.*

**But again, we put them into a global compilation and we loose the context of their study.**

*The Larsen et al. (2014) study is not relevant for our comparison as granitoid soil-mantled hillslopes are considered. This is stated in the text and also discuss lithologic effects in the text associated with Figure 3B. We focus on granitoid sample locations to minimize lithologic effects, comparing a different lithology to our results will of course yield differences. But it is true that a global compilation of data has always its problems. For instance, the compiled SPRs are calculated based on many different parameter assumptions. In addition, the compilation of MAP or MAT from a specific global data set may twist the global compilation in an unknown way. However, neglecting the fact that, a global comparison has its internal limitation, the use of a global data set may also be rewarding (e.g., Mishra et al., 2019). Please also see our response above that we've added extensive text to our methods section on limitations of global compilations.*

**In short, one could argue that much of the story centres on the Heimsath data, and so this not really a "global" analysis.**

*See answer above and addition of Table 2 and improvement of Fig. 4. Other studies and geographic locations at high MAP are also used in this study.*

**Overall, there needs to be a more cautious tone adopted throughout in the use of the published data.**

*The manuscript has been revised to use published data more carefully and also present more information related to the data included. For instance, we included a new Table 2 which contains some information about the sample region or the number of published samples. We hope that the addition of this Table helps to shed light on the use of the published data.*

**3) One aspect of prior work on this theme is not well captured in the manuscript: West et al., 2005, outlined how chemical weathering rates in felsic catchments (derived from river solute**

**fluxes) can be viewed as supply or kinetically limited. Others had proposed such terms, but the paper was perhaps the first to provide the data, framework and empirical models to explain global patterns we see. One of the findings of that work relevant here, is that a link between chemical weathering and climatic variables (MAP, runoff, and/or MAT) should not be expected where denudation rates are low and weathering rates are controlled by supply. In other words, Figure 2 is not surprising for sites where soil has developed and supply may control weathering rates**.

*It is true that West et al. (2005) were one of the first introducing the terms supply- and kinetically-limited chemical weathering rates. We acknowledge this publication in the introduction (and elsewhere throughout the manuscript) when referring to the review work of Riebe et al. (2017), which explains the different frameworks used by geomorphologists and geochemists. According to West et al. (2005) a link between chemical weathering and climate variable should not be expected where denudation rates are low. However, this statement may hold true for river catchments (which West also included in his analysis) but not necessarily in soil-mantled hillslopes, which can be in a transitional state (e.g., Brosens et al., 2020). This caveat is also discussed by West et al (2005) in their publication. Soil-mantled hillslopes in slowly eroding settings with thick soils could be kinetic-limited due to vegetation and biotic weathering factors – we discuss mechanisms for this in the text. We also now discuss in the text the recent work by Oeser and von Blanckenburg (2020 Biogeosciences) who come to a similar finding as our study in that high amounts of vegetation (and NPP) result in intensified nutrient recycling within an ecosystem, and subsequently lower biogenic weathering. Their approach however was very different and relied on 87Sr/86Sr measurements in plants, soil, rock (in the same Chilean study areas). We've highlighted more strongly this study and the similarity in interpretations in this revised version.*

**Other comments:**

**Point 2 in the conclusions - this was not addressed - it seems to be prior work**.

*The handling editor is right that this point was not discussed. The point has been removed from the hypothesis in the introduction as well as in the conclusions.*

**Point 3 in the conclusions - as above, there are 4 sites, yet this is a sweeping statement of spatial patterns across a continent**.

*We state in the conclusions that "The observations made in the Chilean Coastal Cordillera and our comparison to global data lead to the following conclusions for the hypotheses stated in section 1.0". Furthermore, we state that the observations could have several explanations and needs further investigations.*
*The conclusions have been heavily edited to emphasis that the observations from limited data of the Chilean study are also manifested at a global scale when considering similar lithologies.*

**General comment to the handling editor and both reviewers.**

We thank the two anonymous reviewers for their constructive and insightful comments. We appreciate the time they have spent on evaluating this work. Nearly all of the suggested changes are reasonable and we implemented the changes as described in this response to reviews.

General comments were addressed in rewriting chapter 1 Introduction in order to present a coherent text to the reader. In addition, chapter 3 Methodology was adjusted to the suggestion made. Last but not least a chapter 5.3 Study caveats and challenges was added to the discussion. The specific comments are addressed below. The reviewer's comments are in bold, the replies in italics, and changed text in normal font.

**Response to reviewer comments 1 for preprint esurf-2021-22**

**General comments**

**The manuscript presents soil and saprolite data from a climate gradient in Chile in granitic lithology. It investigates the potential controls of climate and vegetation on rates of soil production, total denudation and chemical/physical weathering. In addition, it compares the Chilean data to a global dataset and models of soil production. The manuscript is generally well written, and the discussion of the data is very detailed covering multiple aspects of how climate and biota can potentially affect soil production and related properties. I also very much appreciated the assessment of previously developed soil production models against real data. These parts of the discussion are very commendable.**

*Response: We thank the reviewer for her/his enthusiasm about the manuscript, and the constructive comments that follow. We find all find that all of the suggested changes are reasonable and will lead to an improved manuscript.*

**However, considering the constraints of the Chilean dataset (geochemical variability within sites, limited replication of sites, outlier treatment), there is the real danger of overinterpretation. In context of the global datasets, the interpretability and potential for generalisations (at least for granitic lithologies) improves, but I would still advocate for not overinterpreting the soil production/weathering data as mainly driven by climate/vegetation, since 1) most of the interpretation is based on visual assessments of scatter plots, and 2) other drivers of soil production, like tectonic uplift, should also be considered in the assessment. While the paper is naturally focussed on identifying climate/vegetation as drivers of soil production/weathering, I feel that the discussion does not sufficiently challenge this link. In addition, I highlight a specific issue with an external dataset that gives reason for extra caution when analysing collections of regional case studies. When these regional case studies are subsequently linked to environmental covariates sourced from global models and datasets of coarse spatial resolution (e.g., global climate models, global-scale topographic/vegetation data), mischaracterisation of case study sites can easily occur.**

**In summary, I think there is strong merit in the publication, combining a regional case study with a global perspective on soil production in granitic lithologies, if the constraints in the data and methodology are adequately acknowledged, and a more balanced discussion of the patterns in the data is provided.**

*Response: We thank the reviewer for highlighting this concern. Throughout the manuscript we do mention and discuss various caveats such as the reviewer mentions (e.g., geochemical variability, different rock uplift rates, etc…). However, our approach was to address these factors in a dispersed way throughout the text that apparently diminished the impact / intent for readers (not our intention). Therefore, to add more emphasis and clarity to potential caveats, in the revised manuscript we will add a new discussion section (5.3 Study Caveats and Challenges) at the end of the document that discusses the items mentioned by the reviewer (and others) into one place.*

*Please note that while we agree with the reviewer that geochemical variations and other factors such as geographically different rates of uplift, and resolution of climate and vegetation data used can impact the relationships looked at, there are a few fundamental factors that need to be realized by readers. First, the soil production, denudation, and chemical weathering rates looked at in this study are integrated over thousands to hundreds of thousands of years (i.e. the integration timescale pertinent to interpreting cosmogenic nuclide derived denudation rates). Thus, inherent to our approach is a temporal averaging of results. This means site specific local variations in vegetation or climate are averaged out to some degree. Second, spatial variations in rock uplift rate are indirectly considered in our study via consideration of slope. Topographic slope is strongly dependent on rate of rock uplift (and lithology, which we've accounted for as best as possible by focusing on granitic settings). Third, throughout the study, we focus on the large-scale trends in the data over a range of precipitation rates and vegetation cover. It would be an unusual coincidence if **local** scale variations in chemical composition of bedrock or vegetation / precipitation resulted in a global or even regional (e.g., Chile) trend as we document here. Rather, more likely is that these factors are the cause of the variance in the data along the global trend, rather than the trend itself.*

*In summary, in the new discussion section, we discuss the factors mentioned by this reviewer as well as the potential impact they have on our interpretations.*

**Specific comments**

**L29-43**

**I think this could benefit from a restructure and rewrite. Several concepts are presented (regolith, soil erosion, soil production, soil denudation) but the individual sentences are not well linked up into a coherent line of thought. It reads more like a collection of definitions where the reader has to fill in the gaps but not as an introduction.**

*Response: Sorry about this. The section mentioned was restructured and rewritten with the intention to introduce a coherent line of thought.*

**L62-76**

**Given your study is strongly linked to a previous publication, co-authored by you (Oeser et al 2018) and containing similar/same datasets, can you please indicate clearly in the introduction: what is the novel aspect of this new manuscript?**

*Response: Corrected as suggested.  Thanks..*

**L98**

**This comments also applies to the other site descriptions:**

**I would suggest some clarifications here 1) soil horizon thickness: do you combine A and B horizon thicknesses for this? 2) Clay content, pH, bulk density values: Are these profile averages across all soil horizons?**

*Response: For the clarification of the two points raised by the reviewer we added at the end of Section 2 "Chilean study areas" and before 2.1 "Pan de Azucar" a section indicating in what the different parameters are measured and where the values are reported.  Hopefully, this addition clarifies the manuscript. The added section is:" The combined thickness of A- and B-horizons is considered as soil thickness (see Table S1 in Oeser et a., 2018). The reported clay content, pH, and bulk density are the pedon averages of each study area (see Table 3 in Bernhard et al., 2018). The chemical index of alteration (CIA; after Nesbitt and Young, 1982) for bedrocks is a study area average whereas the CIA for regolith is reported for specific horizons (for more details see Table S5 in Oeser et al., 2018).  The cosmogenic nuclide-derived denudation rates are reported for South- and North-facing mid-slope positions (see Oeser et al., 2018 and Table S6 in there)."*

**L124**

**Regardless of the different ways to define soil, regolith, saprolite etc., an Umbrisol is a soil type and not a regolith type after WRB.**

*Response: In order to avoid confusion due to the different soil definitions, the sentence has been changed to: "The Umbrisol has soil horizons as thick as 60 to 90 cm and a clay content of 26.2 ±2.6%.". We hope that this is correct.*

**L133**

**Since saprolite plays an important role in the manuscript, how was it recognised and distinguished from mobile regolith in the field when sampling?**

*Response: The following clarification has been added to the text in question: "The top of saprolite is considered to be the first encounter of in situ weathered bedrock represented by the C-horizon. This sampling strategy is a common approach for calculation of soil production rates from cosmogenic nuclide measured in pedons (e.g., Dixon et al., 2009). Representative photographs of this horizon from the Chilean study areas are available in Oeser et al., (2018: Figures 3 to 6)."*

**L144**

**What are the uncertainties in Tables S2 to S5? Are they all SEM? Not fully clear in the table.**

*Response: Sorry about that. What the uncertainties represent have now been described in foot notes of Table 2 to 5.*

**L164**

**What is SP$_{soil}$? Do you mean SPR?**

*Response: Yes, we do mean SPR. The oversight has been corrected.*

**L165**

**Are the concentrations of Zr and that of other elements in each soil a weighted average of the entire soil profile (see also the tables in the supplement)? Given that these concentrations can vary significantly throughout the depth profile of a soil, it would be good to clarify what these values that represent each pedon are.**

*Response: yes – the reported values are the average concentrations for all samples in the layer referred to. We have more explicitly clarified this in the text and reads now like:"…., where Zr$_{soil}$ is the average Zr concentration for soil samples from the pedon. Similarly, Zr$_{sap}$ is the average Zr concentration of the saprolite samples from the pedon. Zr$_{rock}$ is based on the average of all bedrock samples collected in one specific study area (see Table S3 based on Oeser et al., 2018)."*

**L174, L62-64**

**I would recommend to directly link the methods in 3.3 to the hypotheses. How are the methods used for testing the individual hypotheses (e.g., correlation estimation between which variables, significance tests). Also, the hypotheses only refer to the Chilean sites, but you appear to test those hypotheses across the global dataset as well. The discussion preamble (L247-249) also does not quite conform to these hypotheses.**

*Response: We have now explicitly stated this link to hypotheses in section 3.3.*

**L178**

**I would recommend making the statistical analysis clearer: you estimated the Pearson correlation coefficient assuming linear relationships between your variables and tested for statistical significance of these relationships (using a t-test I presume, under the assumption of normally distributed data).**

*Response: Thanks for this suggestion. We've clarified the text, and also mentioned that table S8 contains the R2 and P values.*

**L179**

**I would appreciate some more information in the main text on what kind of models these are. The details of each model are well placed in the supplement but some general description of what they are doing would be very helpful to be included in the methods. The would support the understanding of why you include these models in the first place and how this comparison of model predictions and your data contributes to testing the hypotheses.**

*Response: We have now rewritten section 3.3 and present the basic concepts of each model considered in the main text.*

**L194**

**Are the SPR-related uncertainties also SEM?**

*Response: No, they are not. We have now clarified this (see response above) in the corresponding table S2.*

**L232-234**

**Why were other samples that show a negative weathering rate not excluded but only NAPED20?**

*Response: Ooops. Thanks for catching this. The data were not handled consistently. We now include all samples and do not remove any. The text has been adjusted.*

**L238**

**Looking at Figure 2A it seems that La Campana is only different to Nahuelbuta because of the very high value of LCDEP30. This sample was excluded when summarising the data for the La Campana site because of the steep slope (L223, S4). If only this this sample was also excluded from Figure 2 based on its very high values (i.e., outlier because of it unusually steep slope), the differences between La Campana and Nahuelbuta would completely disappear (2A, 2C), and likely won't be statistically significant for any other panel in Figure 2, given the sample**

size and uncertainties. Considering inherent geochemical variability at each site (as reported in the results section and later discussed below) and differences in topography between sites (e.g., effect of slope), how can you be sure that the differences between the sites, particularly between the 2 most humid sites, are indeed mainly a reflection of differences in climate/vegetation and not of other reasons? It is interesting to note that Oeser et al (2018) also considered differences in uplift rate and topography as reasons for the differences between the two most humid sites (6.1.1 in Oeser et al. 2018). As such, statements as in L307-308 sound less convincing, including claiming the "commonalities" with the global dataset that is interpreted as mainly driven by climate/vegetation (L255+).

Also considering my comment on L232-234, there seems to be a lack of consistency in the treatment of so-called outliers. See also L265-266 – an exclusion for which the reason is not well explained in the text (at least to my understanding).

*Response: Thanks for your thoughts on this. It's important to note that with the number of available samples from each Chilean study area (n= 4 to 7) that statistical significance cannot be established as suggested in the comment. Nevertheless, we see your point. We have changed Figure 2 so that the sample locations are color coded by topographic slope (as reported in Table S1). We have also modified the text so to state that the LC study area shown in Fig. 2 is either equivalent the NA area, or potentially higher but that given other differences between the areas this cannot be accurately resolved. As a side point – it is worth remembering that global data set suggest higher values for this precipitation rate.*

**L271**

The maximum SRP's shown in Fig 3B appear to come from Larsen et al (S7). Having some regional knowledge of their sites, the precipitation values derived from Karger et al. are well below the values from actual observations and those of the regional climate models (e.g., https://niwa.co.nz/climate/national-and-regional-climate-maps/west-coast). For instance, the Rapid Creek sites are less than 10 km from a rainfall gauge (Cropp River) that receives >10 m of MAP (https://data.wcrc.govt.nz/cgi-bin/HydWebServer.cgi/sites/details?site=81&treecatchment=3). Using a national NZ dataset (the data is available here: https://data.mfe.govt.nz/layer/53314-average-annual-rainfall-19722013/), the SRP maximum at ~3000 mm would disappear for then non-granite sites in Figure 3B and shift to between 7500 and 8000 mm; by using regional climate data, there would be no evidence for a humped relationship between SRP and precipitation. The rainfall data of the national NZ rainfall model should be better adapted to the extreme orographic conditions of NZ's Southern Alps than a global model. See also the general comments.

While I can't comment on data specifics in the compiled granite dataset given the limited review time, the lack of a similar observation in non-granitic lithologies (as far as presented here) reduces the general application of the observations in granite lithologies and should be reflected in the later discussion, including the following paragraph and 5.2.2.

*Response: Thank you for your thoughts on this. We have modified the text in the caption to mention that higher precipitation rates are documented for NZ from other data sets you mention. However, we think it is more important to handle all data in the same way when comparing to things such as climate data so that there is a consistent handling of not only how the precipitation data is processed, but also the time span of data used. The ERA-interim data is the basis for the CHELSA climatology we use in this study. ERA-interim reanalysis data is a standard and trusted data product for climatological studies. To not use this one region, or two select different 'regional' climate data sets for each study area would not be recommended. There are simply too many differences (methodological and observational) with how climate data can be processes. We chose to avoid a picking-and-choosing of different climate data sets that can introduce biases in to the analysis. However, much of this discussion is irrelevant because we don't actually use the NZ data in our analysis that the reviewer refers to because they do not come from granitic settings.*

*While we appreciate the reviewer's interest and knowledge on this topic, we respectfully disagree with this suggestion on several levels. In the spirit of scientific exchange, we elaborate a bit on this (although please keep in mind we are not even using the NZ data in our analysis). First, in climate literature it is frowned upon to compare a single weather station measurement (as suggested above) to a climatological downscaling result. Point measurements from weather stations frequently disagree with climatologies (which present an spatially and temporally integrated average….which is of higher relevance for comparing to temporally average cosmogenic nuclide SPR data). The average of many weather stations within a 'grid box' of a climate data set would be more appropriate, but not possible in this case. Second, the CHELSA data set (Karger et al., 2017) used here is a 30-arc sec resolution (900 m) which is, for climate data, high resolution. The CHELSA data is a downscaled data product that uses weather station and ERA reanalysis data within it. We couldn't confirm if the station mention by the reviewer is included in it, but given the thoroughness of the ERA data it most likely is if it's over a 30 year time span. One of the advances of the CHELSA data set is also it's consideration of orography and valleys (see Karger et al., 2017). Finally, the CHELSA data set was peer reviewed (Karger et al, 2017) and as far as we could tell, the NZ web sites referred to above are not peer reviewed.*

*In summary, we've added text to the manuscript stating that local meteorological measurements may differ from what is reported here, but that we focus our analysis on a set of consistently processed, peer reviewed, global data to avoid biases in downscaling results between more regional or local studies.*

**L349-354**

**I miss the discussion of tectonic uplift as a potential driver for differences in soil production rates. Could some of the pattern in the SRPs of the global dataset not also be linked to tectonics? I suspect that not all data points in Figure 4 and 5 are subject to similar tectonic uplift rates and this is briefly touched on in L262. Put differently, for given uplift rates, would the same patterns regarding vegetation and climate parameters persist? You have done this**

**for different slope classes in Figure 4, should a similar analysis not also be done for uplift rates?**

*Response: Thank for mentioning this. We've modified the last paragraph of section 5.1 to explain better how different rates of tectonic uplift are manifested in slope angles.*

**L277**

**What does 'This' refer to?**

*Response: Sentence fixed / clarified.*

**L317**

**The comparison with the EEMT approach is only shown in S6 but not discussed in the text. I would recommend discussing them as well to allow for a full cross-model evaluation.**

*Response: We've now clarified this in the text in section 3.3 (Methods). We prefer to not explicitly discuss it within the text because a) we don't want to attack another study in detail in our manuscript; b) the original study of EEMT does not compare to the same breadth of data (only stream data of Riebe et al); and c) the model does not come close to fitting observations presented here …. So we don't see the point in an already long enough paper to discuss a poor fitting model.*

**L366-369**

**This sounds contradictory – first, it is stated that bedrock Zr is lower than soil and saprolite Zr, but then an example is presented, where this is not the case. And there are other examples in the data where Zr in soil or saprolite is lower than in the rock (e.g., see AZPED sites).**

*Response*: We thank you for pointing this out. We have clarified the text and implications for this.

**L374-375**

**The large uncertainties around Zr are acknowledged in the preceding sentences, but then the 50% of CDF (a variable heavily depended on Zr) is rather firmly interpreted as the ceiling for CDF values (for sites "where chemical weathering happens"). I would recommend to word this accordingly to reflect the uncertainty of the Zr data. This also applies to the discussion of the differences in CDF and Wtotal between the two wettest sites (L387-416). It goes to great lengths in explaining the potential drivers behind the differences in the data, but I think it should also be acknowledged that because of the chemical variability and the limited replication, the differences between sites may not be only a reflection of climate/biota but also of other factors. This comment is similar to a previous one regarding Figure 2.**

*Response:  Thanks for the comment.  The text has been modified to mention this and tone it down.*

Schaller and Ehlers combine new and existing measurements and calculations of soil production, chemical weathering, and physical erosion rates to examine trends between climate, vegetation, weathering, and erosion at four sites along a transect of the Andes spanning diverse precipitation, temperature, and vegetation zones. They analyze trends at these four well-characterized sites and extend their analyses to a global compilation of similar measurements in granitic catchments worldwide to demonstrate the nonlinear and non-monotonic relationships between climate, vegetation, weathering, and erosion.  I found the manuscript to be well organized and written, with high quality figures clearly conveying the results. More importantly, I also found the analyses and discussion to be interesting and well conceived — not only to address the hypotheses of the study, but also to examine the complexities within the large and diverse datasets it aggregates.

*Response:  Thank you for your time in reviewing the manuscript and your enthusiasm for it.*

That said, my most substantive comment is that, in a few places (noted below), I felt that greater explanations of how the hypotheses and results presented here relate to previous work are needed to properly put the results in context. For instance, the hypotheses follow a summary of past research in the region that largely contradicts the hypotheses, yet no explanations or references substantiating the hypothesized relationships between soil production, erosion, climate, and vegetation are provided. Of course, these are largely explained (and even demonstrated in the figures later) by comparing the data to various empirical predictions in the results section, but I believe proper context and referencing is needed in the introduction. Moreover, many references in the discussion seemed to point readers to past studies rather than identify and explain the relevant connections between the presented results and this past work.

*Response:  We are a little bit confused on what exactly the reviewer is referring to without an example or two.  Nevertheless, we've tried to clarify things by modifying the last paragraph of the introduction where hypotheses are presented to reference other literature, and also allude to the results we present.  We have also modified different parts of the discussion section, as suggested, to mention when our results agree or disagree with previous studies.*

**GENERAL COMMENTS:**

- I appreciate that paleoclimate considerations are discussed in lines 86-94 and I wonder if other paleo-environmental conditions (and potential changes) that may affect your results and interpretations are worth considering. For instance, I presume measurement of the various factors that may explain the low contribution of chemical weathering to total denudation in the southernmost study site (e.g. solute fluxes, organic acid concentrations, soil thickness, microbial abundance) are based on modern observations, but have these also

**remained constant (or at least similar in pattern amongst the study sites) over the integration timescales of cosmogenic erosion rates?**

*Response: Well…. Very interesting point! Unfortunately – this a large problem facing these types of studies because, with the exception of paleoclimate predictions (which are also often quite sparse) there simply are not data sets available for the other items mentioned like paleo solute fluxes, paleo microbial abundances, etc. However, the point raised is a good one. In the new discussion section 5.3 (Study caveats and challenges), that was added in response to the other reviewer, we have added text addressing this comment and that future studies could also address this as it adds an unknown to our (and every other paper on this topic) analysis.*

**- It seems worth addressing how slopes have been calculated in the studies utilized in the global compilation (e.g. if they have been calculated from similar resolution DEMs and over similar length scales and differencing or averaging schemes e.g. topographic slopes vs steepest-descent slopes)…unless standardized slope measurements from GTOPO30 have been used (column 15 in table S8)? I don't suspect that differences in calculation would dramatically affect the binning into (rather generously sized) 10 deg slope bins, but given the significance of this binning on your analyses and known e.g. scale-dependence of slope measurements, I do think this warrants some discussion.**

*Response: We have now clarified how slopes were computed in the supplemental material where we already discuss this, and also in the figure captions. For brevity – figures NOT involving a correlation analysis use the original reported slope by the study (when available). If the original study did not report a slope the symbols were not color coded to indicate this. For the correlation analysis we wanted to avoid differences between how individual studies calculated slopes and treated all data the same – so for the correlation analysis we extracted the slopes form GTOPO30.*

**MAIN TEXT LINE COMMENTS:**

**Throughout: numerous places where north and south are unnecessarily capitalized (e.g. in "North- and South-facing" and e.g. line 256). Clauses beginning with "which" should be preceded by a comma.**

*Response: Thanks. Fixed. The document was scanned for North and South and changed to north and south where necessary. In addition, commas were added to clauses beginning with which.*

**Line 20: right parenthesis missing at end of sentence**

*Missing parenthesis was added.*

**Line 42: I am a bit confused here if denudation rate refers to chemical denudation rate specifically or to total denudation rates (guessing the latter)…particularly since "total denudation rates" are referenced in lines 55-56. Perhaps add "total" before denudation rate or "(physical plus chemical denudation)" after?**

*Response: Yes, we meant total denudation rates and corrected the section in question.*

**Lines 62-64: I'm a bit confused about the hypotheses since they seem to partly disagree with the previous observations just discussed…? Perhaps this should be explained briefly (or appropriate citations added supporting these ideas)?**

*Response: Thank you for highlighting this. We've done a massive rewrite of the introduction to make it clearer and have specifically set up more clearly the contradictions in previous work and how our hypotheses stem from them.*

**Lines 80-81: Minor point, but I'd suggest rephrasing "where the neighbouring…" since I think "due to subduction of the Nazca Plate" does not fully explain the along-strike similarity to which you are referring (the Nazca Plate also subducts below the Northern Andes, where you have flat slab subduction) and it's worth noting that the tectonic regime within the study regions is similar beyond just Nazca Plate subduction**

*Response: Rephrased to state similar geometry and orientation of the subducting Nazca plate near the coast.*

**Line 83: delete comma**

*Response: Comma was removed*

**Line 88: delete comma**

*Response: Comma was removed*

**Line 90: exist → existed**

*Response: Error was corrected*

**Line 109-110: add comma after 43 and delete comma after citation**

*Response: Comma was added respectively removed*

**Line 151: missing right parenthesis after units of soil production rate. Add "are" before "the mean…"**

*Response: Missing parenthesis was added and the "are" inserted.*

**Line 152: length → lengths**

*Response: S was added*

**Line 157: Is there some new meaning to the square brackets used to enclose variable units? If not (as I presume), I'd suggest standardizing throughout**

*Response: In physics / geophysics literature units are typically (not always) reported in square brackets. We changed it to be consistent throughout the manuscript.*

**Lines 178-180: Seems like this sentence should only state that the leaf area index LAI and SPRs at sample locations were compared to model predictions (if I understand correctly)**

*Response: The sentence was corrected as suggested.*

**Line 232: missing right parenthesis after first "yr"**

*Response: Parenthesis was added*

**Line 240: Why do you only point out the similarity in rates between La Campana and Nahuelbuta for physical erosion rates? The majority of the rates also appear to agree between the two sites for soil production and chemical weathering, no?**

*Response: Thanks for catching this. We've modified the other relevant parts of the text to address this.*

**Line 247: you've been using an oxford comma up to here…add comma after "weathering" for consistency**

*Response: Comma was added*

**Lines 273-275: Optional, but it would be helpful to briefly summarize e.g. Heimsath and Whipple's finding about the influence of lithology and rock strength variations on SPRs here and move the citation to the end of the sentence, instead of simply directing readers to that paper**

*Response: The reference has been added to the end of the sentence.)*

**Line 277-278: Couldn't it alternately suggest that MAP simply influences SPR non-monotonically, even in the absence of other processes/environmental differences? I'd at very least suggest rephrasing "this observation could suggest that…" Perhaps "processes" should also be changed to e.g. "factors" since MAP is not a process.**

*Response: Sentence changed as suggested.*

**Lines 279-280: move citations to end of sentence**

*Response: Citations were moved to the end of sentence*

**Line 280: "relationship in" → "relationship between"**

*Response: Suggested change was made*

**Line 289: weekly → weakly**

*Response: Correction was made. Thank you.*

**Line 290: "suggest an even weaker correlation" → "correlate even more weakly"**

*Response: Sentence was changed as suggested*

**Line 291: missing oxford comma**

*Response: Missing comma was added*

**Line 295-296: "hillslopes with slopes >30 deg…decrease with decreasing slope"?? I'm very confused about what this refers to since Figure S4 appears to show increasing trends with slope. Perhaps this should be a citation to Figure 4 and say "SPRs within the highest slope bins decrease with slope"?**

*Response: Latter half of sentence reworded to clarify meaning.*

**Line 298: I'm not sure it's necessary to say "qualitatively" since you are showing a quantitative trend here. Suggest deleting.**

*Response: "Qualitatively" was deleted*

**Line 299: where by → whereby, SPR →SPRs**

*Response: Both errors were corrected*

**Line 308-309: I find "SPR is a process of chemical weathering and physical erosion" to be a bit awkwardly worded. Perhaps change "SPR" to "soil production" or "SPR depends on both chemical weathering and physical erosion"?**

*Response: Sentence corrected as suggested.*

**Line 317: Should Figure 5B be referenced here too?**

*Response: Text was rearranged and reference is not needed anymore.*

**Lines 322-323: These two sentences confused me, since the black bold line in 5A shows the empirical prediction, no? I suggest rephrasing to clarify this (perhaps "…are predicted to increase rapidly with…"**

*Response: Corrected as suggested by the reviewer.*

**Lines 349-350: suggest changing "a combination of increasing…vegetation" to "variations in MAP, soil depth, and vegetation worldwide" since your data and the model-data comparisons show that non-monotonic relationships and co-variation of these factors can perhaps explain the global variations**

*Response: The sentence was changed as suggested and reads now as:"* In summary, based on the previous considerations, we find that global variations in observed SPRs can be explained by variations in MAP, soil depth, and vegetation worldwide (Fig. 5)."

**Line 354: occurs → occur, "for different slope areas observed" → "across different hillslope gradients" or simply "across the different slopes observed" perhaps (in any case, I find "slope areas" to be confusing since it could be confused for some metric of drainage area, too)**

*Response: Sentence was adjusted and reads now as:"* We note that although high-slope settings produce the highest SPRs, the trends in MAP and vegetation causing an increase and then decrease in SPRs as MAP or LAI increase occur across different hillslope gradients (Fig. 4). "

**Lines 366-367: I think this statement should be qualified a bit ("may still be meaningful") since the extent to which the measured Zr concentrations of bedrock truly reflect the Zr concentration of the parent rock from which soil and saprolite derived is still uncertain, even if the sign of change is correct**

*Response: Section has been reworded to address this comment.*

**Lines 368-369: suggest combining these sentences "…is negative because the ZR concentration in the saprolite is lower than…"**

*Response: Sentence was corrected as suggested and reads now as:"* In contrast, the calculated fraction of weathering in saprolite for sample NAPED20 in Nahuelbuta is negative because the Zr concentration in the saprolite is lower than the concentration in the bedrock. "

**Line 370: "over" → "to"**

*Response: Corrected as suggested.*

**Line 371: "as, for instance, in..."**

*Response: Corrected as suggested.*

**Line 375: add comma after "happens"**

*Response: Corrected as suggested.*

**Line 378: "orth" → "north"**

*Response: Corrected as suggested.*

**Line 379: "neither" → "not"," "nor" → "nor by"**

*Response:  Corrected as suggested.*

**Line 389-390: by effectively diluting it? Perhaps change "diminish" to "dilute" if so...?**

*Response: "diminish" was replaced by "dilute" as suggested.*

**Line 398: "leading to an" → "which may underestimate Wtotal and overestimate Esoil"**

*Response: Sentence corrected as suggested and reads now as:"* Due to this possible Zr loss in regolith, the calculated $W_{soil}$ is a minimum value which may underestimate $W_{total}$ and overestimate $E_{soil}$.*"*

**Line 403: "As with" → "With"**

*Response: Corrected as suggested.*

**Line 406: the passive voice here makes it unclear if the attribution of the stabilizing effect of plants on the decrease in physical erosion rates has been proposed previously in other studies (guessing not?). "We attribute the..." or perhaps "The decrease in physical erosion rate may result from..."**

*Response: Sentence corrected and reads now as:"* The decrease in physical erosion rate may result from stabilizing effects of plants.*"*

**Lines 407-408: "increases" → "increase", "is lower again" → "decrease again"**

*Response: Sentence was adjusted as suggested.*

**Line 410: suggest flipping sentence "Precipitation, temperature, and pH all affect microbial abundance"**

*Response: Sentence corrected and reads now as:"* Precipitation, temperature, and pH all affect microbial abundance (e.g., Fierer and Jackson, 2006; Bahram et al., 2018; Tan et al., 2020).*"*

**Line 412: "in" → "along", "increase" → "increases", "decrease" →"decreases"**

*Response: Sentence was corrected.*

**Line 414: perhaps this statement should be qualified "which may explain the decreasing pH values from north to south and the lower bacterial abundance…"**

*Response: The corrected sentence reads now as:"* A comparable increase in MAP and decrease in MAT is observed in the Chilean Coastal Cordillera, which may explain the decreasing pH values from north to south and the lower bacterial abundance in Nahuelbuta than La Campana (Bernhard et al., 2018; Oeser et al., 2018).*"*

**Line 415: "made observations are unique" → "hypotheses are valid"**

*Response: Sentence was changed to address the comment.*

**Line 420-421: I think this sentence should be split to clarify that the observed corrrelations between denudation rates and chemical weathering do not derive from this same global compilation.**

*Response: Sentence fixed/clarified.*

**Line 425: "is absent or reduced" → "is not operative or occurs only at low rates"**

*Response: Sentence was adjusted.*

**Line 430: "increasing" → "to increase", "diminishing" → "diminish"**

*Response: Adjusted as requested.*

**Line 431: "not only do climate and vegetation…, but so does topography"**

*Response: Sentence was corrected.*

**Line 443: delete "setting with"**

*Response: Deleted.*

**Line 450: missing right parenthesis at end of sentence**

*Response: Missing parenthesis added.*

**Line 458: add "monotonically" since they do increase with MAP at lower MAP values**

*Response: Suggested correction made.*

**Line 459: "vegetation" → "vegetation cover"**

*Response: Missing word added.*

**Line 459-460: I think this citation needs to be explained more clearly/thoroughly - particularly as it related to trends between chemical weathering rates (and SPRs) and MAP and/or vegetation**

*Response:  Section in question was reworded.*

**Line 471: I'm not sure "stabilize" sufficiently/properly describes the trend. "increase and then stabilize"**

*Response: Sentence corrected as suggested.*

**Line 474: "the low contribution of chemical weathering to total denudation"**

*Response: Corrected as suggested.*

**Line 475: "where solute fluxes are high and soils and saprolites are rich in organic acids"**

*Response: Corrected as suggested.*

**Line 478: I'd suggest also adding "and non-monotonically"**

*Response: Suggestion was incorporated.*

**Line 640: "bin slopes" → "slope bins"**

*Response: Corrected as suggested.*

**Line 641: "two-polynomial" → "binomial"**

*Response: Changes was integrated.*

**Line 647: 'for different mean annual precipitations and zero soil depth and blue lines are for different MAP and soil depths, assumed to covary (Supplemental Text 2)" Is that correct or is one MAP assumed for the blue lines?**

*Response: Fixed.  Panel B was adjusted to panel A and figure captions adjusted.*

**Line 652: Pluralize all (or none) of the y-axis variables**

*Response: Corrected as suggested.*

**Figure 2 - Optional, but it seems like with both different marker colors and symbols you could also display slope aspect and relative elevation information cited in the text (which might help illustrate the trends you discuss)**

*Response: Yes – we color coded the symbols by slope.*

**SUPPLEMENT LINE COMMENTS:**

**Line 57: delete extra right parenthesis**

*Response: Deleted.*

**Eq. S4: EMT → EEMT**

*Response: Corrected.*

**Lines 60-61: I'm guessing the S5s and S6s here should be changed to S3 and S4?**

*Response: Corrected.*

**Figure S3: What are the dashed lines in A?**

*Response: these dashed lines are the 95% confidence level.  We added this to the figure caption.*

---

## Author Response (AR3)

**General comment to the handling editor**

We thank the handling editor for his additional insightful comments which are addressed below.  In the track changes version of the manuscript, we show the differences between the last submission and our revisions. The comments of the handling editor are in bold, the replies in italics, and changed text in normal font.

Best wishes,
Mirjam Schaller (corresponding author)

**Comments to the author**:
Dear Authors,

First, please accept my apologies for the delay. This was related to my recent move to a new institution, which resulted in a delay due to ESurf emails not making it to the new address, and then a delay related to my own workload. I am sorry.

Thank you for your efforts in revision and replying to the reviewer comments and my own.

Having read through in detail these responses, I am satisfied that the major issues have been accounted for with revisions throughout the manuscript.

However, the new version has become a little more wordy, and there were some repeated clauses and tangential information. In addition, there was still one example of overreaching in the final section. I have edited the PDF and make suggestions for a final round of minor revisions to help address the first round of review comments.

Please let me know if any comments or suggested edits are unclear.

Best regards,
Bob Hilton
AE ESurf

**Response to comments of the associate editor Bob Hilton**

Lines 46-49:

**I suggesting mentioning these concepts, but I wonder if the text to follow would be better linked to the discussion in the previous paragraph on what these terms mean for weathering, and also to seed that soil studies can be helpful. something like: "Supply limited landscapes are those where increases in erosion can increase chemical weathering, while kinetically limited captures locations where other drivers (most commonly temperature, precipitation, fluid residence time) become important. While these concepts have proved useful, they may not well explain heterogeneties at the hillslope scale. In addition, complimentary to the previous work, other studies… etc,."**

*Text generally changed to the suggestion of the associate editor with minor changes. The text reads now like:* "Supply-limited landscapes are those where increases in physical erosion can increase chemical weathering, while kinetically-limited landscapes capture settings where other drivers (most commonly temperature, precipitation, fluid residence time) become important. While these concepts have proved useful, they may not well explain heterogeneities at the hillslope  and catchment scale. In addition, complimentary to the previous work, other studies…

Line 74:

**New paragraph here**
*New paragraph inserted as recommended.*

Line 76:

**delete "taken together", a repeat of previous sentence**
*"Taken together" has been deleted.*

Lines 101-102:

*Sentence deleted as suggested.*

Lines 104-108:

**To address the previous two aims, our efforts are focused on evaluating two hypotheses. : 1) soil production as well as chemical weathering rates increase with increasing MAP rates.  Norton et al. (2014)  Perron (2017)  White and Blum (1995); and 2) the contribution of chemical weathering to total denudation is constant over a climate gradient.  Riebe et al. (2004a) and Larsen et al. (2014).**

*The deletions as suggested have been accommodated. The section reads now like:* "To address the previous two aims, our efforts are focused on evaluating two hypotheses: 1) soil production as well as chemical weathering rates increase with increasing MAP rates (e.g., Norton et al., 2014; Perron, 2017; White and Blum, 1995) and 2) the contribution of chemical weathering to total denudation is constant over a climate gradient (e.g., Riebe et al., 2004a; Larsen et al., 2014)."

Lines 211-212:

**This was unclear as written.**

*Sentence changed to improve the understanding of the text:*" As a result, the SPRs reported here and the total denudation rates in Oeser et al, (2018) are the same expect for La Campana and Nahuelbuta where Oeser et al. (2018) reported rates based on cosmogenic production rates corrected for vegetation cover.*".*

Lines 215-217:

**To evaluate these hypotheses, , SPRs from the Chilean study…**

*Suggested deletions addressed. The sentence reads now as:* "To evaluate these hypotheses, SPRs from the Chilean study areas were compared to previously published SPRs derived from granitoid (Table 2 and S5) and non-granitoid (Table S6) soil-mantled hillslopes from around the world."

Line 222:

**As per my review comment, please note (something like):**
**In addition, some studies have specifically sort out the highest SPRs to assess the potential limits to soil production (Heismath Nat Geo, Larsen Science)**

*The suggested sentence was added to the text:* " In addition, some studies have specifically sort out the highest SPRs to assess the potential limits to soil production (e.g., Heismath et al., 2012; Larsen et al., 2014*).*"

Lines 222-228:

**To avoid oversimplifying our comparison of the Chilean data to other globally distributed studies, we have  selected previous studies reporting data most akin to our sampling and analysis approach.  from soil-mantled 225 hillslope measurements rather than catchment average estimates from river load, and also locations with granitoid lithologies underlying the hillslopes to minimize lithologic variability effects. **

*Changes and deletions were addressed as suggested:* "To avoid oversimplifying our comparison of the Chilean data to other globally distributed studies, we have selected previous studies reporting data most akin to our sampling and analysis approach from soil-mantled hillslope measurements rather than catchment average estimates from river load, and also locations with granitoid lithologies underlying the hillslopes to minimize lithologic variability effects."

Lines 232-234:

**The global data sets used here for LAI and climate therefore benefit from having the same, consistent, processing of data, but suffer from having a coarser resolution than local to regional based studies**
*Sentence was deleted as suggested.*

Line 238:
**spell out why this is a potential issue - something like:**
**which means chemical weathering may not be as responsive to changes in climatic and biotic drivers.**
*Sentence adjusted to suggestion:* "The potential bias introduced by this is that these areas may be supply-limited which means that chemical weathering may not be as responsive to changes in climatic and biotic drivers."

Lines 241-243:
* * *
*Sentence deleted as suggested.*

Line 244-250:
**This paragraph contains some repeated info, and I wonder if the tables can be mentioned in the previous paragraph and the methods briefly too, thus this paragraph deleted.**
*The paragraph was shortened and coupled to the next paragraph. We hope that this change is acceptable. The new paragraph reads now like:* "For the Chilean and global data sets our analyses were conducted in two ways. First, the SPRs and the compiled topographic, climate, and vegetation parameters for granitoid sample locations were analyzed with a Pearson linear correlation analysis (Table S7). Also, the compiled chemical weathering and physical erosion rates determined in granitoid soil-mantled hillslopes around the world (Tables 2 and S8) were compared to climate and vegetation parameters based on a Pearson correlation analysis (Table S9). In addition (second), we compare...."

Lines 264-273:
**~~In this section, results for soil production, chemical weathering, and physical erosion rates are presented for the new and previously published cosmogenic nuclide concentrations from the Chilean study areas (Fig. 2; Tables S1, S3, and S4). Results are given for each study area starting in the arid north and progressing to the south. The total denudation rates (Dtotal) presented below are the composite of the total chemical weathering rate (Wtotal) and the physical erosion rate (Esoil) and based on the calculated SPRs and the Zr concentrations in rock, saprolite, and soil (Table S2). The previously published total denudation rates (Oeser et al., 2018) were recalculated to account for chemical weathering of saprolite (see methods). Because the observations presented in our global compilation were previously published (see references in Table 2), we do not present the compilation in this section but rather in section 5 (Discussion) where~~**

**it is integrated and discussed in the context of our Chilean observations. In addition, correlations of SPRs as well as chemical weathering and physical erosion rates with parameters are also shown in section 5.**
*Paragraph deleted as suggested.*

Line 364:
**This is where you need to mention what the reviewer flags - the highest SPRs at 3000 mm/yr may have higher precipiations.**
**You can say what you want about model vs measurements, but I'm sorry, it rains a lot in the western Southern alps, and a lot more than 3000 mm/yr.**
**So your text needs to be fair to that, rather than just saying there is an up and down again. Because if you put those samples at higher precipitation, clearly it doesn't support your claim.**
*We have modified the text to add this caveat as you suggest. The text now says:*
*"The maximum SPRs in these settings are not only higher than in granitoid lithologies but also reached at a higher MAP (~3,000 mm/yr). However, we note that the SPRs coming from these high MAP settings are located on the western flank of the Southern Alps, New Zealand where other studies (e.g., Larsen et al., 2014), using different climate data, have suggested higher precipitation rates (>3,000 mm/yr). We caution that if MAP is higher than 3,000 mm/yr in this location, then the functional relationship between SPR and MAP in granitoid vs. non-granitoid hillslopes is not the same as suggested here."*

Lines 487-489:
**repeated information from a few lines previous?.**
*Sentence in question deleted.*

Lines 552-555:
**this discussion seems out of scope of the manuscript**
*Two sentences in question deleted.*

Lines 552-555:
**This paragraph contains too much hyperbole as highlighted in the review process. The Chiean sites are 4, and have complex patterns. Therefore to start the whole discussion with "it is astonishing that the global dataset shows a similar picture…" is too much.**
**Also the important discussion has already been made in the two previous sections**.
*We have deleted text at the start and end of the paragraph that could be considered 'hyperbole' (including the sentence mentioned above). We also restate that four Chilean study areas were investigated. The rest of the paragraph was kept because it makes a comparison of our work to previous published work which and sticks to observations presented in those study. The remaining text and numbers cited are not covered elsewhere in the manuscript and we think it is important to reference related, previous studies (most journals request a section with 'comparison to previous work'). Hopefully you find this acceptable.*
*Please note – we modified the paragraph to be more cautiously worded and use expressions like "if this similarity in limited observations is not coincidence, then …" and "If true, then these observations [previous work we cite] suggest ….". Thus, we try to make it abundantly clear to readers what the assumptions are in our and others work and that the ideas presented warrant additional observational investigations.*